# SARS-CoV-2 infection dynamics in Denmark, February through October 2020: Nature of the past epidemic and how it may develop in the future

**Steen Rasmussen** [1,2] *, **Michael Skytte Petersen**[3], **Niels Høiby**[4,5]

**1** Center for Fundamental Living Technology (FLinT) Department for Physics, Chemistry and Pharmacy, University of Southern Denmark, Odense, Denmark, **2** Santa Fe Institute, Santa Fe, New Mexico, United States of America, **3** Center for Biosecurity and Biopreparedness (CBB), Statens Serum Institut, København, Denmark, **4** Department of Clinical Microbiology, Rigshospitalet, Copenhagen, Denmark, **5** Institute of Immunology and Microbiology, Panum Institute, University of Copenhagen, København, Denmark

* steen@sdu.dk

**Data Availability Statement:** All relevant data are within the manuscript and its Supporting

## Abstract

### Background

Initially, the relative sizes of the asymptomatic and the symptomatic infected populations were not known for the COVID-19 pandemic and neither was the actual fatality rate. Therefore it was not clear either how the pandemic would impact the healthcare system. As a result it was initially predicted that the COVID-19 epidemic in Denmark would overwhelm the healthcare system and thus both the diagnosis and treatment of other hospital patients were compromised for an extended period.

### Aim

To develop a mathematical model, which includes both asymptomatic and symptomatic infected persons, for early estimation of the epidemic's course, its Infection Fatality Rate and the healthcare system load in Denmark, both retrospectively and prospectively.

### Methods

The SEIRS (Susceptible—Exposed—Infected—Recovered—Susceptible) model including deaths outside hospitals and separate assessments of symptomatic and asymptomatic cases (based on seroprevalence) with different immunological memories. Optimal model parameters are in part identified by Monte Carlo based Least Square Error methods while micro-outbreaks are modeled by noise and explored in Monte Carlo simulations. Estimates for infected population sizes are obtained by using a quasi steady state method.

### Results

The calculations and simulations made by the model were shown to fit with the observed development of the COVID-19 epidemic in Denmark. The antibody prevalence in the

information files as well as in URLs provided in the references section.

**Funding:** The authors received no specific funding for this work.

**Competing interests:** The authors have declared that no competing interests exist.

general population in May 2020 was 1.37%, which yields a relative frequency of symptomatic and asymptomatic cases of 1 to 5.2. Due to the large asymptomatic population, the Infection Mortality Rate was only 0.4%. However, with no non-pharmacological restrictions the COVID-19 death toll was calculated to have more than doubled the national average yearly deaths within a year. The transmission rate $\Re_o$ was 5.4 in the initial free epidemic period, 0.4 in the lock-down period and 0.8–1.0 in the successive re-opening periods through August 2020. The large asymptomatic population made the termination of the epidemic difficult and micro-outbreaks occurred when the country re-opened. The estimated infected population size July 15 to August 15 was 2,100 and 12,200 for October 1–20, 2020.

## Conclusions

The results of the model show, that COVID-19 has a low Infection Fatality Rate because the majority of infected persons are either asymptomatic or with few symptoms. A minority of the infected persons, therefore, requires hospitalization. That means that for a given infection pressure of both symptomatic and asymptomatic infected there will be a lower pressure on the capacity of the health care system than previously predicted. Further the epidemic will be difficult to terminate since about 84% of the infected individuals are asymptomatic but still contagious. The model may be useful if a major infection wave occurs in the autumn-winter season as it could make robust estimates both for the scale of an ongoing expanding epidemic and for the expected load on the healthcare system. The simulation may also be useful to evaluate different testing strategies based on estimated infected population sizes. The model can be adjusted and scaled to other regions and countries, which is illustrated with Spain and USA.

## Introduction

COVID-19 was a new infection that initially hit the Chinese city Wuhan. Therefore, it took some time before its severity and pandemic properties were realized by China, WHO, CDC and ECDC. The Chinese outbreak was reported to WHO December 31, 2019 and its Emergency Committee declared a Public Health Emergency of International Concern January 30, 2020 and a global pandemic situation March 11, 2020 [1] The international spread of the COVID-19 pandemic was facilitated by the Chinese New Year celebration, which took place January 15—February 11. During this period millions of Chinese people traveled inside China and from abroad e.g. Europe and North America to China to visit their families and then returned after the end of the New Year celebration [2]. Wuhan locked down January 23, 2020 and the rest of China subsequently followed that decision [3].

Many Chinese stay and work in large European cities. One of these cities is Milan, Lombardy, Italy south of the Alps. This was the first region in a western country to be heavily hit by the pandemic officially beginning February 21, 2020 [2, 4]. In February, many schools in Europe closed for one week for winter holiday over which thousands of adults and children traveled from all over Europe to the Alps for skiing.

The first case of COVID-19 reported in Denmark came from Northern Italy February 27, 2020 and subsequently 139 Danes came home after ski-holiday in Northern Italy and Austria,

mostly from Ischgl [5], where they had contracted COVID-19 during the school holiday period, which in Denmark ended February 23. The epidemic in Denmark (5.82 mil. inhabitants) developed rapidly and Statens Serum Institut (SSI), the Danish national center for disease control, estimated that the transmission rate per infected person, $\Re_0$ was 2.6 and the prevalence increased exponentially until March 11, when WHO declared that the COVID-19 was a pandemic and Denmark closed down in the following days [6]. The adjusted information from SSI released June 11, 2020 indicates that altogether 1,488 persons contracted the infection abroad over the first month of the Danish epidemic, initially mostly from Austria and Northern Italy. The exact number of infected that started the Danish epidemic is not clear, so after reviewing the data we decided to approximate the initial infection number to be 690 per February 24, 2020, which is also the number we use as initial conditions in our simulations [7, 8].

COVID-19 was initially imported from skiing areas in the Alps, but then local spread took place inside Denmark so the national strategy changed from prevention and containment to mitigation March 11, 2020 [8], and testing for SARS-CoV-2 was restricted to patients who needed treatment in hospitals mainly because of a severe shortage of testing equipment. After April 1st the national testing strategy was changed and more people were gradually tested leading to more information about the spread of the epidemic. This change to a more comprehensive testing strategy gradually increased and became stable between April 21 and May 17, 2020 [7].

The policies for closing down Denmark were fully implemented by March 17, 2020 and soon thereafter the number of new hospitalized COVID-19 cases decreased dramatically. The observed number of hospitalized patients already peaked April 7–8 (535 total hospital patients $H_{tot}$, of which 146 were patients in *ICU*s). Thereafter the numbers gradually decreased.

After Easter a gradual re-opening of Denmark began April 20, 2020, and continued with fixed intervals of 2–4 weeks until June 8, 2020. Altogether, by August 31, 2020 there had been a total of 17,084 SARS-CoV-2 positively tested persons, 3,031 of whom were hospitalized (~17.7%) and 416 of those patients were treated in *ICU*s (~13.7%), while 625 infected died (~3.7%). As of August 31st, 2020 there were 20 hospitalized including four in *ICU*s [8].

There were five local outbreaks in Denmark between April 10 and August 31, 2020, with the last two starting around August 1, 2020. One outbreak was at a meat processing plant in the city of Ringsted where the infection was introduced by a Polish worker who then mainly infected other Polish workers. Approximately 150 workers from the meat processing plant were infected before the outbreak was eliminated by mid August. The other outbreak was in the city of Århus and was associated with a soccer game July 26th and associated celebrations, a university weekend seminar attended by 80 students, the Muslim Eid festival July 30—August 3, and a funeral of a Somalian rap-musician, which was attended by 500 people. Some of the infected unfortunately included bus drivers that spread the infection to neighboring regions. About 2/3 of the Århus cases were among people originating from Somalia and Palestine and the majority of the infected were young people. The highest incidence per 100,000 inhabitants in Århus during the outbreak was 99 and the outbreak was conquered by the end of August [8].Since we now have more information about the nature of COVID-19, we have developed a mathematical model and a corresponding simulation of the Danish epidemic. We calibrate the simulation using the official retrospective Danish hospital, ICUs, death data as well as a national antibody test conducted late May 2020. Our simulation is used to investigate the relative frequency of symptomatic versus asymptomatic infected, their relative infectiveness, $\Re_0$ during the pandemic, Infection Fatality Rate, as well as the connection to antibody development among the infected. The model is applicable elsewhere in other regions and countries [9, 10].

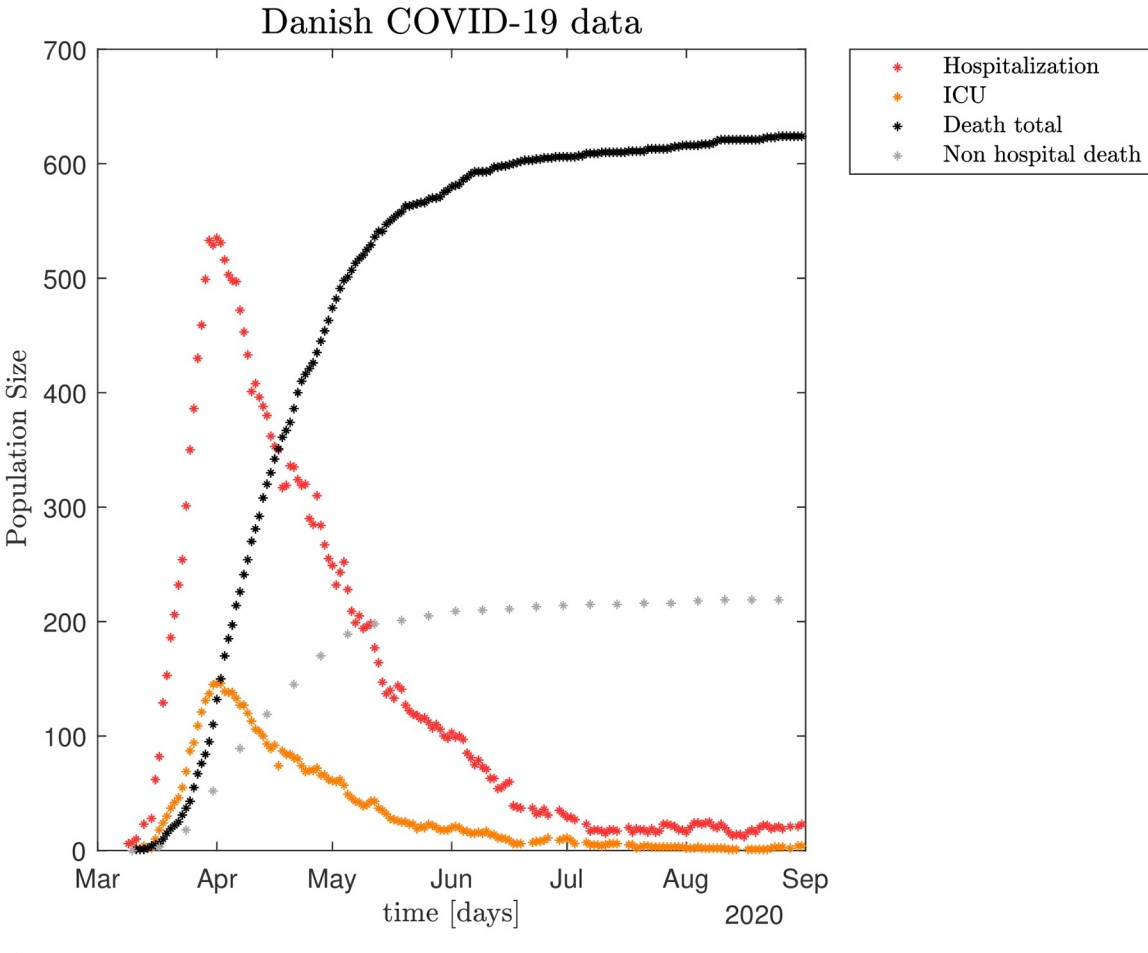

**Fig 1.**

## Material and methods

### Main data sources

Total hospital *Htot* and *ICU* occupation, non-hospital and total death toll numbers, all published daily by SSI [7, 8]. The time-series of these data are shown in Fig 1. Daily hospital admission data, count of PCR positive individuals both outside and inside of the hospital system to estimate the transmission contact number (reproduction number), also called "contact number", $\Re_t$ [8]. For details see S1 File. The mortality of COVID-19 is calculated as Case Fatality Rate (CFR), which is the proportion of deaths from COVID-19 compared to the total number of individuals diagnosed with the disease for a particular period. The mortality is also calculated as Infection Fatality Rate (IFR), which is the proportion of deaths among all infected individuals, including symptomatic and asymptomatic and undiagnosed subjects (e.g. based on sero-prevalence studies).

### The epidemic model system

Our basic SEIRS (Susceptible—Exposed—Infectious—Recovered—Susceptible) style model assumes a single population that can be in one of three infectious states: incubation $I_i$, asymptomatic $I_a$ and symptomatic $I_s$. The $I_i$ state is analogous to the Exposed state in the classical

SEIR model although infection is also possible from the $I_i$ state in our model formulation. The population starts from being susceptible $S$ after transitioning through incubation $I_i$ into one of the two infectious states $I_a$ and $I_s$ and ends up as either recovered $R$ or dead $D$. All three of the infectious states $I_x$; $x = i, a, s$ can reduce the susceptible population $S$ by transitioning members thereof into $I_i$.

A small fraction of the recovered $R$ slowly moves back to the susceptible population $S$ as only a time limited immunity is assumed, which we refer to as the population's average immunological memory. Since there likely is a significant difference in the immunological memory for the symptomatic and the asymptomatic infected we have disaggregated the recovered population $R$ into $R_a$ and $R_s$ with short and long immunological memory respectively. The rate of recovered population members once again becoming susceptible is given by $\xi_y R_y$; $y = a, s$, which is reflected in the positive terms in the first expression in Eq (1).

During the infection a fraction of the symptomatic population $I_s$ can become seriously ill and are either transferred to a hospital $H$ or an $ICU$. From the regular hospital unit $H$ patients can recover, move to $ICU$, or die $D$. Patients from $ICU$ can either move to $H$ or die $D$. Seriously ill individuals can also die at home or in an eldercare facility. These non-hospital deaths are indicated by $M$, where terminally ill patients arrive directly from $I_s$. A flow diagram of the infection and healthcare dynamics is shown in Fig 2. The differential equations that define the flow in Fig 2 are given in Eqs (1) and (2).

The epidemic dynamics is defined by SEIRS model as follows:

$$\frac{dS}{dt} = -(\beta_i I_i + \beta_a I_a + \beta_s I_s)S/N + \xi_s R_s + \xi_a R_a$$

$$\frac{dI_i}{dt} = \frac{(\beta_i I_i + \beta_a I_a + \beta_s I_s)S}{N} - \gamma_i I_i \tag{1}$$

$$\frac{dI_a}{dt} = (1 - \rho_s)\gamma_i I_i - \gamma_a I_a$$

$$\frac{dI_s}{dt} = \rho_s \gamma_i I_i - \gamma_s I_s$$

$$\frac{dR_a}{dt} = \gamma_a I_a - \xi_a R_a$$

$$\frac{dR_s}{dt} = (1 - h_{frac} - m_{frac})\gamma_s I_s + \rho_{h,r}\gamma_h H - \xi_s R_s$$

where the parameters are further discussed in Table 1. Negative terms in the above equations indicate loss of respective population members with time, while positive terms indicate population member growth. The equation for the change in the susceptible population $S$ describes how susceptible are infected from three different populations $I_i$, $I_a$, and $I_s$ with different rates $\beta_i$, $\beta_a$, and $\beta_s$ normalized by the population size $N$. The fraction of incubated that moves from $I_i$ to $I_s$ is defined by $\rho_s$ and the fraction of incubated that moves from $I_i$ to $I_a$ is therefore $1 - \rho_s$. A discussion of the size of $\rho_s$ can be found in S1 File.

Recovered individuals slowly becomes susceptible again as they lose their immunity, where we assume a longer immunity time of $1/\xi_s$ days for the recovered symptomatic $R_s$ and a shorter

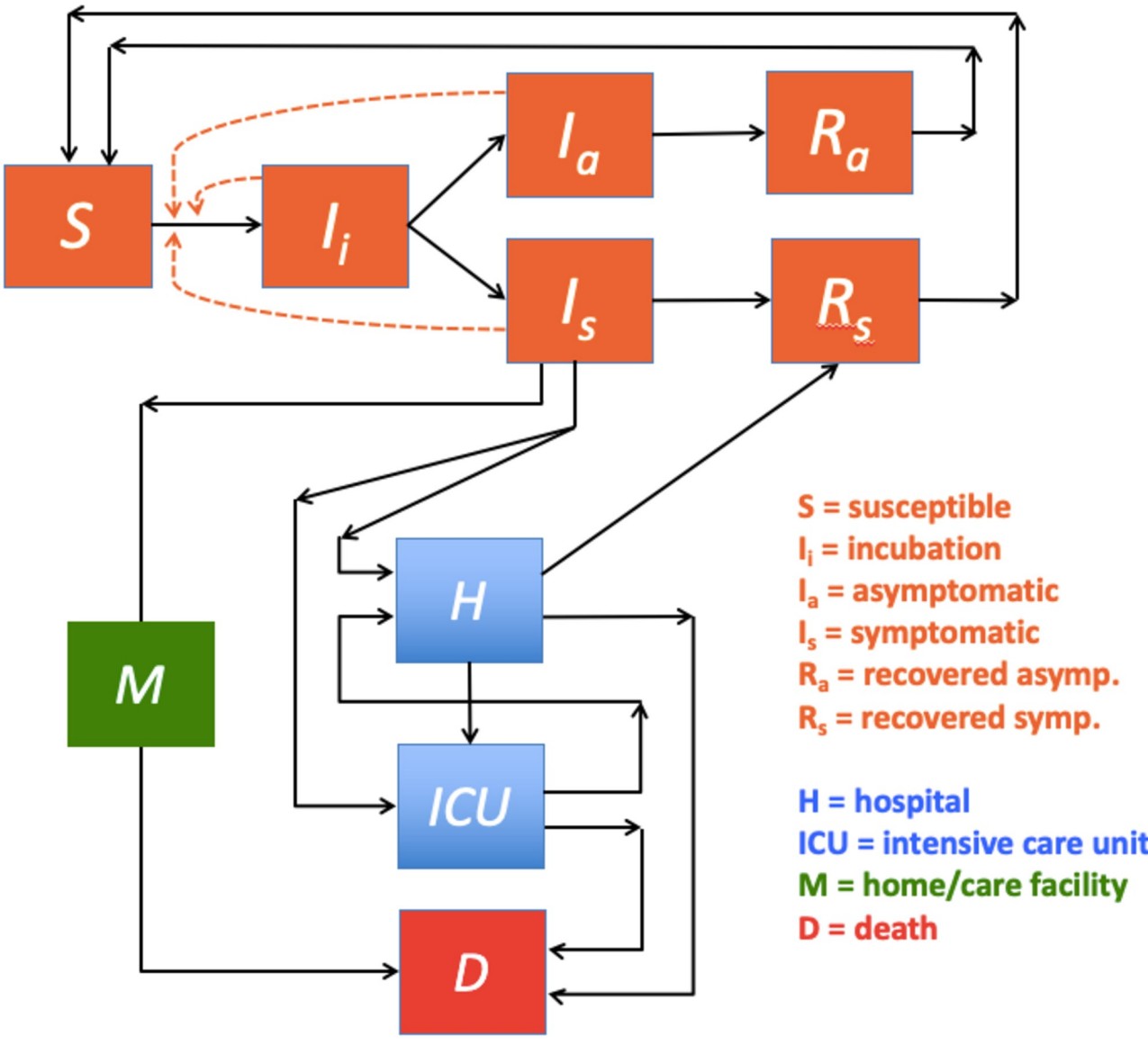

**Fig 2.**

immunity time of $1/\xi_a$ days for the recovered asymptomatic $R_a$. The newly infected are moved from $S$ to $I_i$, the incubating population where they on average reside for $1/\gamma_i$ days. On average $1 - \rho_s$ of the incubated $I_i$ becomes asymptomatic $I_a$ where they on average reside for $1/\gamma_a$ days. Also, $\rho_s$ of the incubated $I_i$ becomes symptomatic $I_s$ where they on average reside for $1/\gamma_s$ days. A further discussion of the immunological memory can be found in S1 File.

A small fraction $h_{frac}$ of the $I_s$ population becomes severely ill and are hospitalized, see Eq (2), while another small fraction $m_{frac}$ becomes so ill that they are not moved into the hospital system but stay at home or in a home care facility before dying, see Eq (2). All asymptomatic individuals $I_a$ recover and are moved into $R_a$ where we assume they retain some immunity for an average of $1/\xi_a$ days. Finally, most symptomatic infected recover $R_s$ directly from $I_s$, as well as from the hospital system $H$, see Eq (2), where we assume they retain some immunity for an average of $1/\xi_s$ days.

**Table 1. Table with input parameters to the combined SEIRS, hospital and home care model.**

| Parameter List | | | |
|---|---|---|---|
| **Parameters** | **Definitions** | **Default values** | **References** |
| $\beta_i$ | incubation infection rate | variable | SE |
| $\beta_a$ | asymptomatic infection rate | variable | SE |
| $\beta_s$ | symptomatic infection rate | variable (1.09/day) | SE |
| $\gamma_i$ | 1/(incubation period) | 1/(5 days) | (1) |
| $\gamma_a$ | 1/(asymptomatic period) | 1/(10 days) | (1) |
| $\gamma_s$ | 1/(symptomatic period) | 1/(10 days) | (1) |
| $\rho_s$ | symptomatic fraction | variable (0.16) | SE |
| $\xi_a$ | 1/(immunological memory for $R_a$) | 1/(60 days) | see section II C |
| $\xi_s$ | 1/(immunological memory for $R_s$) | 1/(700 days) | see section II C |
| $h_{frac}$ | $I_s$ fraction hospitalized including $ICU$ | variable (0.0820) | (2) |
| $m_{frac}$ | $I_s$ fraction terminal ill non-hospitals | variable (0.0081) | SE |
| $\alpha$ | fraction to $H$ | 0.85 | SE |
| $\rho_{h,r}$ | $H$ fraction to $R$ | variable (0.775) | SE |
| $\rho_{h,icu}$ | $H$ fraction to $ICU$ | variable (0.135) | SE |
| $\rho_{icu,h}$ | $ICU$ fraction to $H$ | 0.72 | (3) and SE |
| $\rho_{h,dh}$ | $H$ fraction to $D_H$ | variable (0.09) | SE |
| $\gamma_m$ | 1/(non-hospital terminal ill period) | 1/(8 to 10 days) | SE |
| $\gamma_h$ | 1/(hospitalization period) | 1/(8 days) | (3) and SE |
| $\gamma_{icu}$ | 1/($ICU$ period) | 1/(10 days) | (3) and SE |

SE = simulation estimates. The table parameter values, sometimes in parentheses, refer to a "standard run". It will be indicated in the text if different parameters are used. Note that $\rho_{h,r} + \rho_{h,icu} + \rho_{h,dh} = 1$.

The dynamics of the hospital including its *ICU* and their deaths, as well as the non-hospital death dynamics are defined as follows:

$$\frac{dH}{dt} = \alpha h_{frac}\gamma_s I_s + \rho_{icu,h}\gamma_{icu}ICU - \gamma_h H$$

$$\frac{dICU}{dt} = (1-\alpha)h_{frac}\gamma_s I_s + \rho_{h,icu}\gamma_h H - \gamma_{icu}ICU \qquad (2)$$

$$\frac{dD_{tot}}{dt} = \rho_{h,d}\gamma_h H + \left(1 - \rho_{icu,h}\right)\gamma_{icu}ICU + \gamma_m M$$

$$\frac{dM}{dt} = m_{frac}\gamma_s I_s - \gamma_m M$$

$$\frac{dD_m}{dt} = \gamma_m M$$

where the parameters are further discussed in Table 1. Severely ill symptomatic individuals enter the hospital system at a rate given by $h_{frac}\gamma_s I_s$ where a fraction $\alpha h_{frac}\gamma_s I_s$ goes directly to the regular hospital unit while $(1 - \alpha)h_{frac}\gamma_s I_s$ directly enters into an *ICU*. The size of $h_{frac}$ is critical for the scaling of the pandemic as the hospital data are the central empirical data by which the simulation is adjusted. We use the age disaggregated data for hospital admissions from Ferguson et al. 2020 [11] together with the actual age distribution within Denmark to estimate an

expected aggregated $h_{frac}$, see S1 File for details. The regular hospital $H$ also receives a certain fraction $\rho_{icu,h}$ of the improved patients from the $ICU$ as expressed by $\rho_{icu,h}\gamma_{icu}ICU$, while patients from $H$ leave the hospital after an average period of $1/\gamma_h$ days. Patients from $H$ either recover into $R_s$, or become more ill and move into an $ICU$, or die and move into $D_{tot}$. The $ICU$ population, as mentioned above, receives $(1 -\alpha)h_{frac}\gamma_s I_s$ directly from $I_s$ as well as $\rho_{h,icu}\gamma_h H$ patients from $H$. $ICU$ patients leave the population with the rate $\gamma_{icu}ICU$ and either improve and move to $H$ with the already mentioned rate $\rho_{icu,h}\gamma_{icu}ICU$, or die with the rate $(1 -\rho_{icu,h})$ $\gamma_{icu}ICU$. The accumulated death toll stems from the hospital rate as $\rho_{h,d}\gamma_h H$, from the $ICU$ rate as $(1 -\rho_{icu,h})\gamma_{icu}ICU$ and from the non-hospital (and non $ICU$) associated rate as $\gamma_m M$. Finally, the symptomatic population that eventually dies in a non-hospital location $M$ is given by $m_{frac}\gamma_s I_s$, with a sick period of an average $1/\gamma_m$ days. The non-hospital death tally is accounted for separately on a weekly basis in $D_m$, while they also are counted in the total death toll $D_{tot}$.

## Infection parameters and relative frequency of symptomatic and asymptomatic infected

With $\rho_s$ we express the relative frequency by which an individual moves from the incubation state to the symptomatic state, while $\rho_a = 1 -\rho_s$ expresses the frequency of movement to the asymptomatic state. We may now define the relation between the infection parameters $\beta_i$, $\beta_a$, $\beta_s$ in the SEIRS model as follows:

$$\beta_a = \zeta\beta_s \tag{3}$$

$$\beta_i = (\rho_s\beta_s + (1 - \rho_s)\beta_a)\tau \tag{4}$$

$$\beta_i = \beta_s(\rho_s + (1 - \rho_s)\zeta)\tau \tag{5}$$

assuming that $\beta_i$ and $\beta_a$ can both be defined relative to $\beta_s$ and $0 < \zeta \leq 1.0$ since we assume the asymptomatic are always less infectious than the symptomatic infected. The value of $\beta_i$ in Eq (4) is defined as the weighted sum of individuals that eventually become symptomatic and asymptomatic respectively, and where $\tau$ may be viewed as the fraction of the incubation time they are infectious. If we e.g. assume the incubating individuals are infectious the last 1.5 days (0.3) of the average 5 day incubation time (22) we get

$$\beta_i = (\rho_s\beta_s + (1 - \rho_s)\beta_a)0.3 \tag{6}$$

We may use clinical data to estimate a reasonable $\zeta$ value. Clinical investigations indicate the viral load in saliva from symptomatic and asymptomatic is comparable and that the peak viral load in saliva is found the last day of the incubation time [1, 12–15]. Symptomatic children and adults have the same amount of SARS-CoV-2 in the upper respiratory tract [16].

From [17] it is further assumed that the relative infectiousness for the last day of pre-symptomatic (incubated that turns symptomatic) can be set to 1.00, while the average of the severe symptomatic, the weak symptomatic and the asymptomatic can be set to 0.89, 0.44 and 0.11 respectively during their infection time. In our study we do not distinguish between weak symptomatic and asymptomatic so the corresponding numbers for our model are 1.00, 0.89 and ((0.44+0.11)/2 = 0.275). We can now adjust Eqs (3) and (4) so that they satisfy these

relative infectiousness numbers

$$\beta_a = \zeta\beta_s \tag{7}$$

$$\zeta = \frac{0.275}{0.89} = 0.309 \tag{8}$$

$$\beta_i = \frac{\beta_s}{0.89}\left(\rho_s\frac{0.5+1.0}{5} + (1-\rho_s)\frac{0.16+0.32}{5}\right) \tag{9}$$

where $\beta_s$ and $\rho_s$ are defined as earlier, and we assume an incubation time of 5 days. We use Eqs (7)–(9) for our standard simulations. The main difference between this infection model and the simpler infection model given by Eqs (3)–(6) is a slightly higher level of relative infectiousness for the incubating population for standard parameters. Simulation experiments with both types of infection models convinced us that we could obtain better fits with data if we adopt a higher weight to the incubating population. This is particularly clear when investigating the shape of the initial infection peak in the hospital data. See S1 File for more details.

As neither the proportionality factor $\zeta$ between $\beta_s$ and $\beta_a$ nor the frequency $\rho_s$ of symptomatic versus asymptomatic in Eqs (4) and (9) are well known (October 2020), the initial part of this work is to explore the epidemiological impact of the relationship between $\zeta$, $\beta_s$, $\beta_a$ and $\beta_i$ as well as $\xi_s$, $\xi_a$ and $\rho_s$ under different assumptions. Note that both infection models Eqs (3)–(6) and Eqs (7)–(9) assume a free epidemic. Policy interventions may be implemented by modifying the appropriate $\beta$ parameters.

The mathematical models defined in Eqs (1) and (2) have a variety of parameters, where some are well defined, e.g. from clinical data, while most are restricted by the necessity for the simulations to reproduce the measured antibody data, the historical hospital and *ICU* occupation data as well as the death data both from the hospital system and elsewhere.

All data fittings of model parameters are based on available data from the onset of the Danish epidemic through August 31, 2020. Further discussions of the pandemic in September and October 2020, as well as hypothetical scenarios further into the future are all based on these estimated model parameters.

The relationship between the infection parameters $\beta_i$, $\beta_a$, and $\beta_s$ is based on clinical studies, recall Eqs (3)–(6) and Eqs (7)–(9). The value of $\rho_s$, the fraction of symptomatic infected, is mainly determined by the observed serological antibody response, which was conducted May 8–28, 2020, and published in early June 2020 [8], together with the average antibody decay-times $1/\xi_a$, $1/\xi_s$ from the asymptomatic and symptomatic infected respectively.

The average antibody decay times $1/\xi_a$, $1/\xi_s$ are estimated based on existing knowledge about SARS and other corona viruses. Despite these restrictions a few parameter combinations still have some degrees of freedom, in particular $h_{frac}$ [11] and $m_{frac}$ that scale the size of $I_s$ and thus the whole pandemic. Also, it has been difficult to obtain explicit data for the fraction of the seriously ill that are directly admitted to a hospital $H$ ($\alpha$) versus *ICU* ($1-\alpha$), as well as $\rho_{h,icu}$ and $\rho_{h,d}$ the fraction that are moved from a hospital to an *ICU* and the fraction that dies from a hospital without going to *ICU* respectively.

Our current study seeks to provide useful information about "What if?" scenarios, including potential infection wave scenarios assumed to follow the well-known winter influenza patterns. Such a model is presented in Fig 2 and Eqs (1) and (2). Further, by adding noise to the mean field approach defined in Eqs (1) and (2), we can also interpret the impact of localized micro-outbreaks of the pandemic, as well as other fluctuations found in the empirical data. Finally, our study also seeks to obtain a better understanding of the relationship between the

critical parameters $\beta_s$ (and thus $\Re_0$), $\zeta$, $\rho_s$ and the $\xi_s$, $\xi_a$ pair that together characterize the COVID-19 pandemic.

## $\Re_0$ for the SEIRS model

The basic reproduction number $\Re_0$ can be derived from our SEIRS model system, recall Eq (1), and is given below. See S1 File for details.

$$\Re_0 = \frac{\beta_i}{\gamma_i} + \frac{(1 - \rho_s)\beta_a}{\gamma_a} + \frac{\rho_s \beta_s}{\gamma_s} \qquad (10)$$

Here $\beta_x$ is the average infection parameter for each of the infected subpopulations $I_x$, $x = i,a,s$; $\gamma_x$ is one divided with the average infection period for each of the infected subpopulations, and $\rho_s$ is the fraction of the incubating that becomes symptomatic. In the following sections we shall, when appropriate, estimate the numerical value for $\Re_0$ during the different stages of the Danish epidemic. For details, see S1 File.

## Lockdown and re-opening of the country

March 11, 2020 the Danish Prime minister announced that the country would be closing down in the following days, which meant that all non-essential professional activities would be shut down, social distancing (2 meters) and enhanced hand sanitation were imposed, only essential shopping was recommended (food and pharmacy), at most 10 individuals were allowed to gather at the same time, movement between different parts of the country was discouraged, international borders were closed, but face masks were not made mandatory. Further, any symptomatic individuals with influenza-like symptoms were highly encouraged to self-isolate and seek medical advice and everybody who had been in contact with an infected were also highly encouraged to self-isolate. Therefore, in the following we assume $I_s$ remains at the low level ~1% of the free pandemic value, both during lockdown and after the country reopens.

We model the closing and the reopening of the country in a similar manner, i.e., by initially decreasing $\beta_a$ and $\beta_i$ during the closedown dates followed by increasing $\beta_a$ and $\beta_i$ the dates where the national reopening policies change. The reopening started April 20 with the reopening of schools for the youngest kids (K-5) together with a number of small businesses including dentists, hairdressers and other businesses where services are rendered and where close physical proximity between a provider and a costumer is necessary. Later reopening activities were implemented May 7, 20, June 8, and August 14, 2020 [18].

We model the closedown process of the country as a sigmoid function for the infection parameters $\beta_s$, $\beta_a$, and $\beta_i$ over a period of approximately 7 days, see S1 File for details. We model the reopening process as a slow (linear) increase of $\beta_a$ and $\beta_i$ over that interval, se S1 File for details. A graphical depiction of the closedown and reopening dynamics is shown in Fig 3. For more details see S1 File.

## Results

### Identifying of infection parameters

The typical infection dynamics and corresponding hospital and non-hospital dynamics generated by the SEIRS model, see equation system (1) and (2), are discussed in Figs 4 and 5. For this simulation the fraction of symptomatic infected is $\rho_s = 0.16$ and the infection model is defined by Eqs (7)–(9). Parameter settings for the infection model are shown in S1 File. These parameters are selected by visual inspection of simulation output ensuring an infection scale

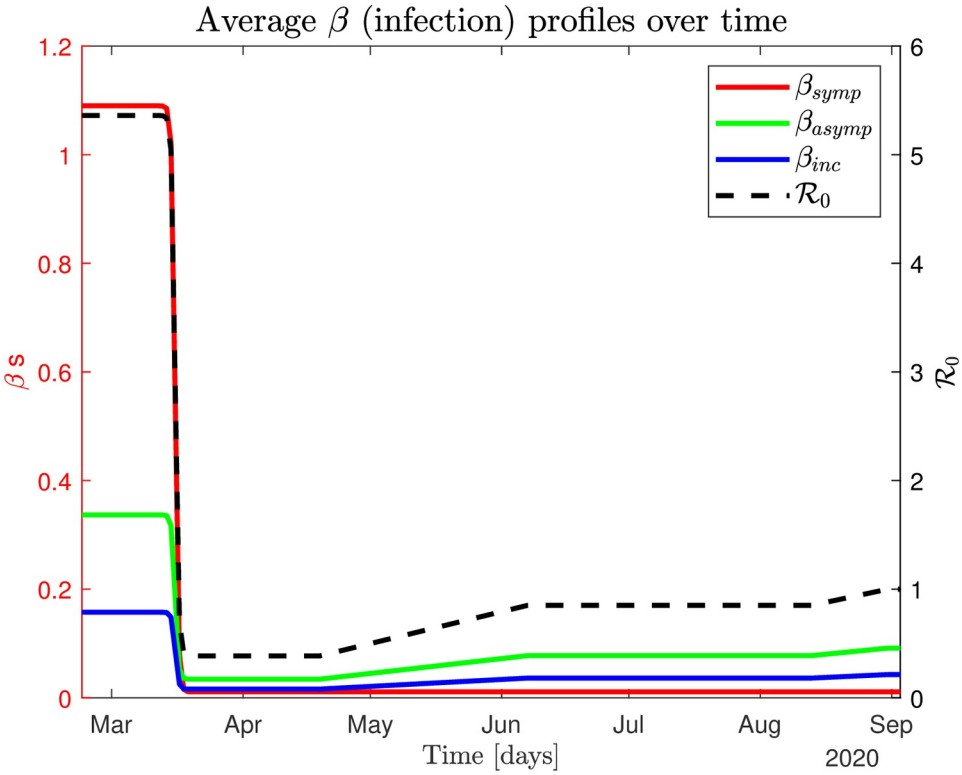

**Fig 3.**

such that ~ 79,700 individuals had been infected by May 28, 2020 [8]. Alternatively, we can utilize a Monte Carlo based Least Squares (MC-LS) optimization method to identify the different $\beta_s$, $\beta_a$ and $\beta_i$. The results from this procedure is shown in Fig 6, and the found parameter values are shown in S1 File. The hospital occupation data has a "shoulder" after the peak (Fig 12 in S1 File),. This means there is an extended period in mid April 2020 with close to constant hospital admissions. Thus, the MC-LS algorithm selects a $\beta_s^0$ that underestimates the observed impact on the health care system as the LS error becomes smallest if the simulated hospital curve is close to as many observed data points as possible. Therefore, the MC-LS estimate misses the full scale of the hospital occupation peak. As a consequence of the initial underestimation of $\beta_s^0 = 1.0583$, the MC-LS algorithm appropriately compensates by slightly increasing the estimate for $\beta_a^{lockdown} = 0.1366 \times \beta_a^0$. Compare Figs 5 and 6.

## Impact of noise

Yet another way to interpret the Danish COVID-19 pandemic data is to view the macroscopic infection dynamics as described by the SEIRS model and the microscopic events as superimposed noise. Thus, the noise can be interpreted as microscopic, localized infection events. We may assume the impact of noise in the hospital data only becomes visible around April 1, as singular microscopic events would be difficult to distinguish earlier because of the dominating macroscopic growth of the pandemic given by $\beta_s^0$ and the successive macroscopic disruption of the infection dynamics due to the sudden lockdown of the whole country. Inspecting the data from April 1 to August 31 supported by $\Re_t$ [8] measured from both hospital admissions and observed infected numbers, yields five distinct peaks corresponding to microscopic events,

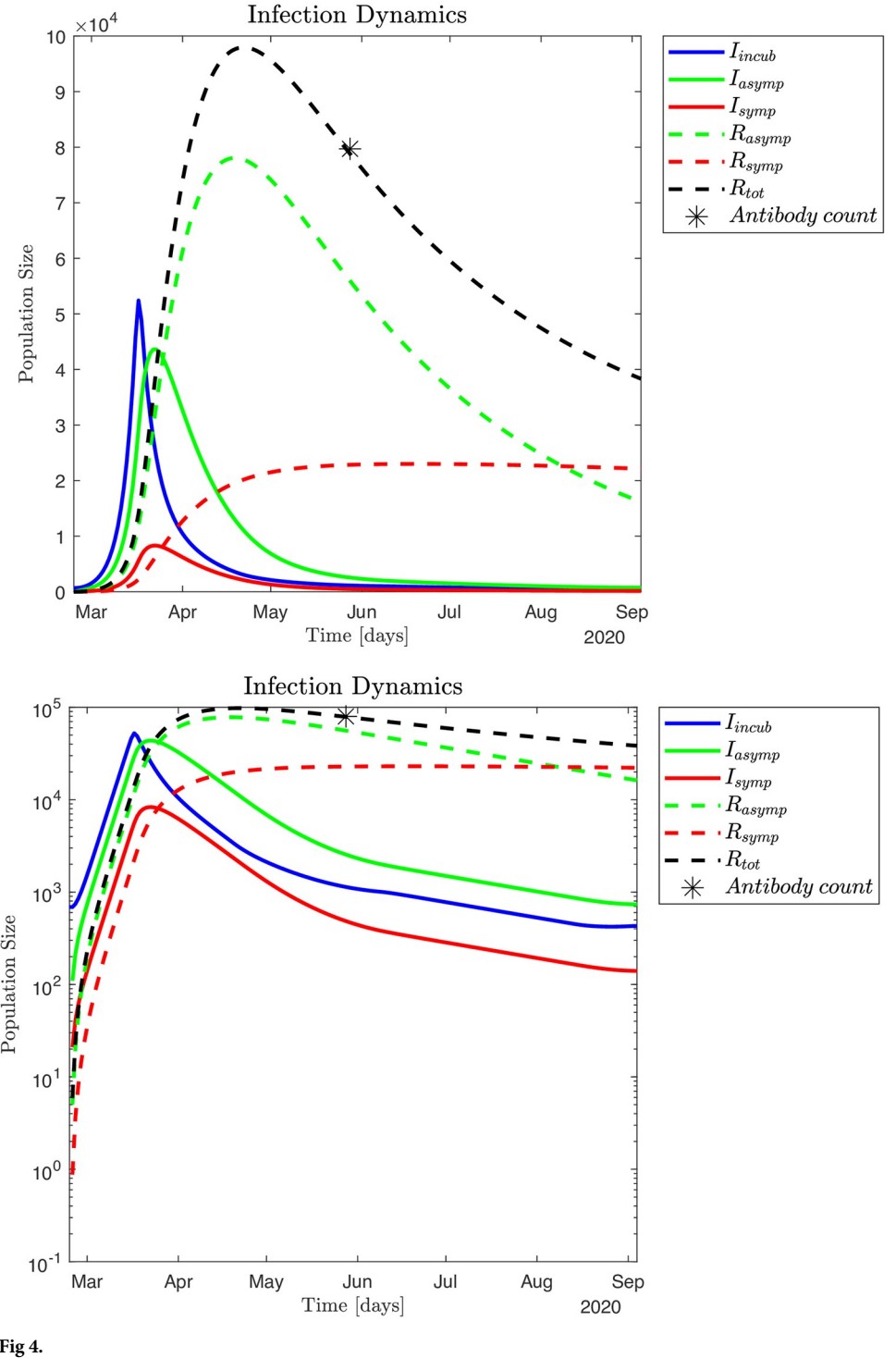

**Fig 4.**

(S12(5,6) Fig in S1 File). Thus, we may estimate *f* as 5 events/150 days = 0.033 events/day. Fig 7 shows 100 Monte Carlo realizations of the epidemic with noise added micro-outbreaks with a frequency of 0.033 events/day and an amplitude of size $A(t)$ = 2,500 for April 1–20, and 1,000 after April 21, 2020. For further details on the scale of the local outbreaks, see S1 File.

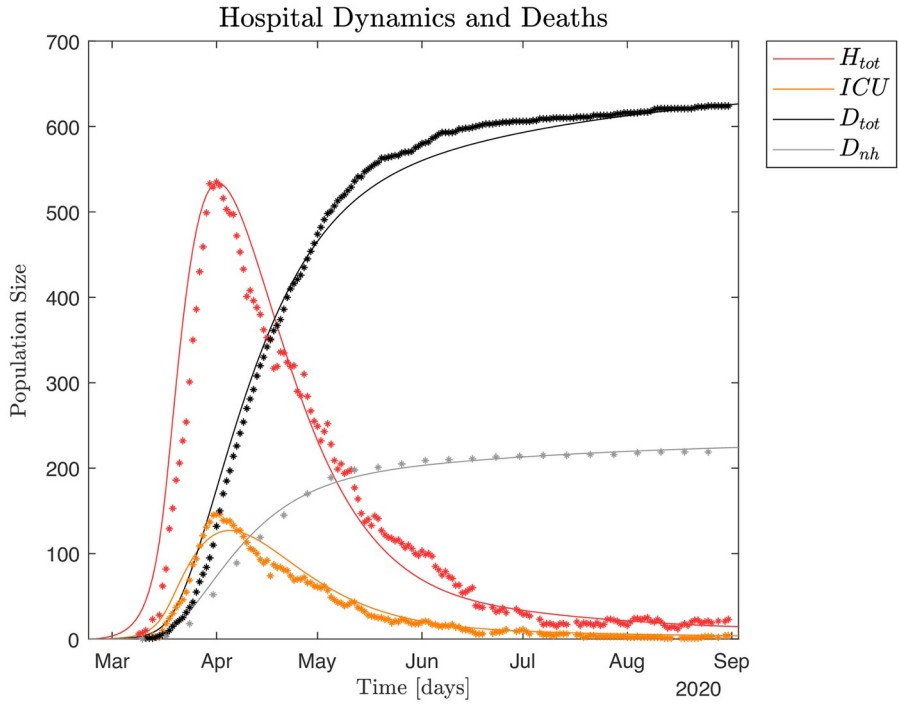

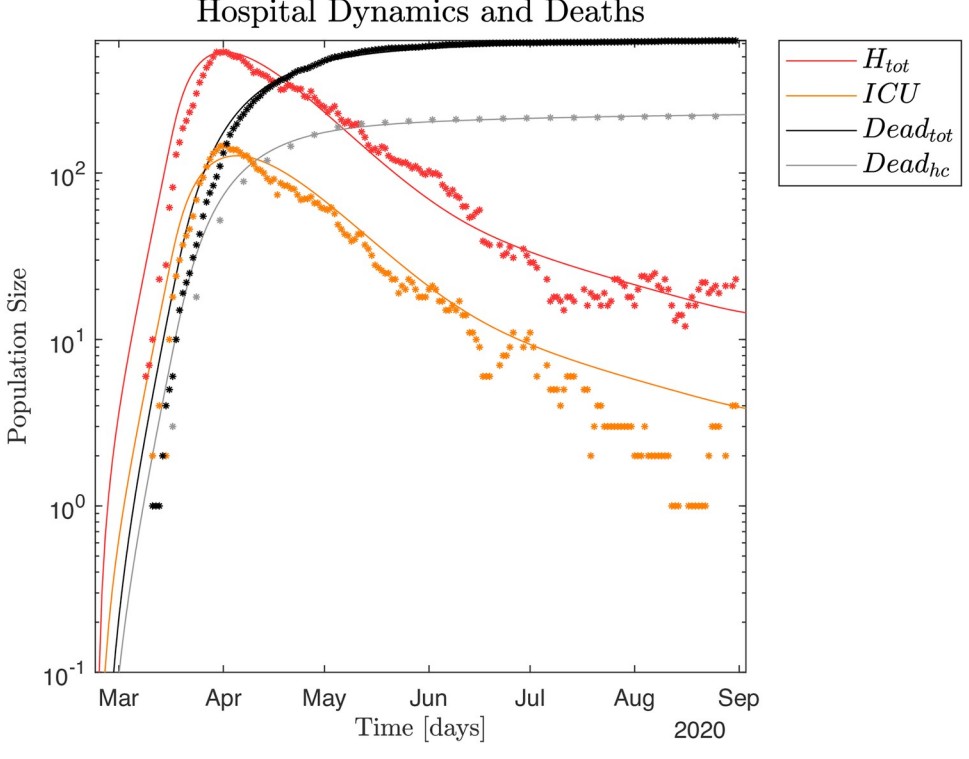

**Fig 5.**

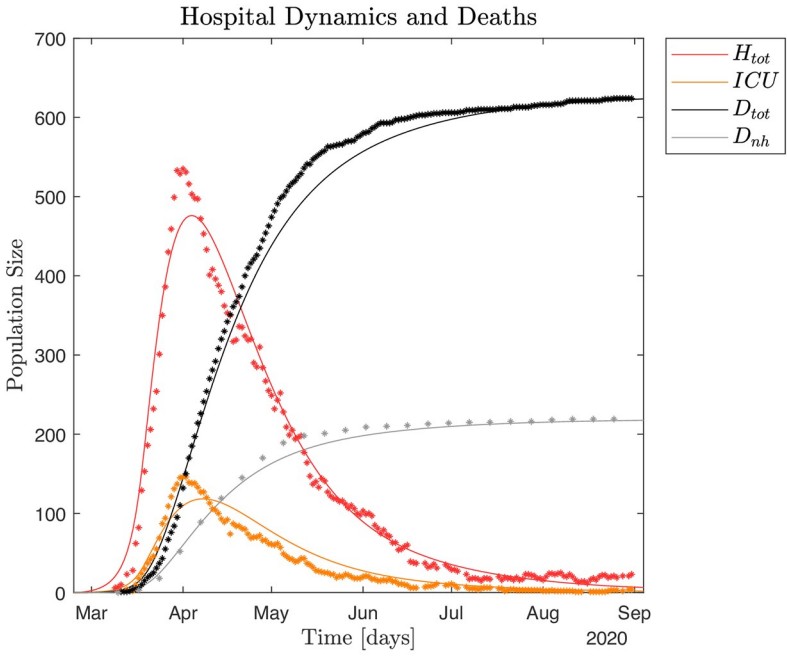

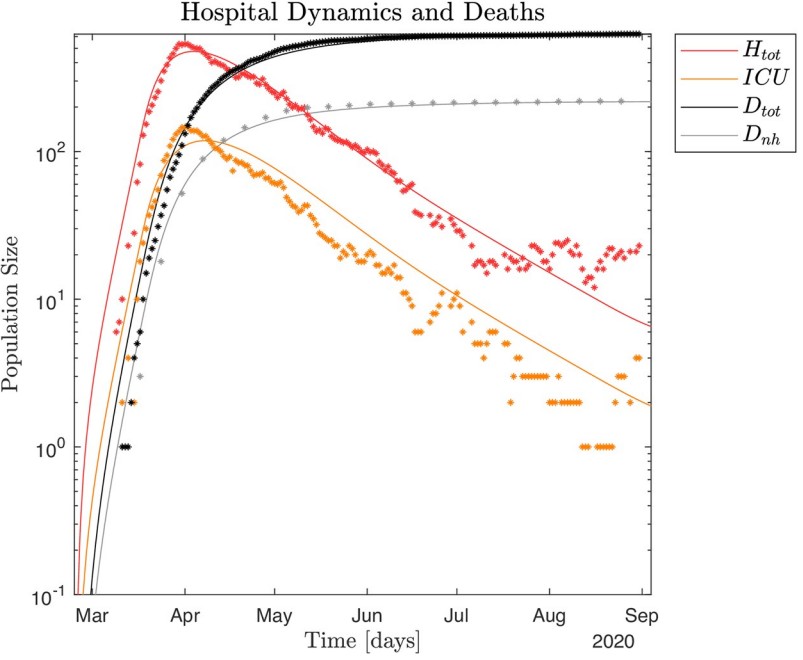

**Fig 6.**

## Symptomatic versus asymptomatic populations

In the current study we define the symptomatic infected population as consisting of individuals with serious symptoms, while the asymptomatic population for simplicity includes individuals with no symptoms as well as weak symptoms, e.g., minor symptoms from upper airways (nose, throat) and no fever. Thus, in our model a symptomatic infected individual knows that

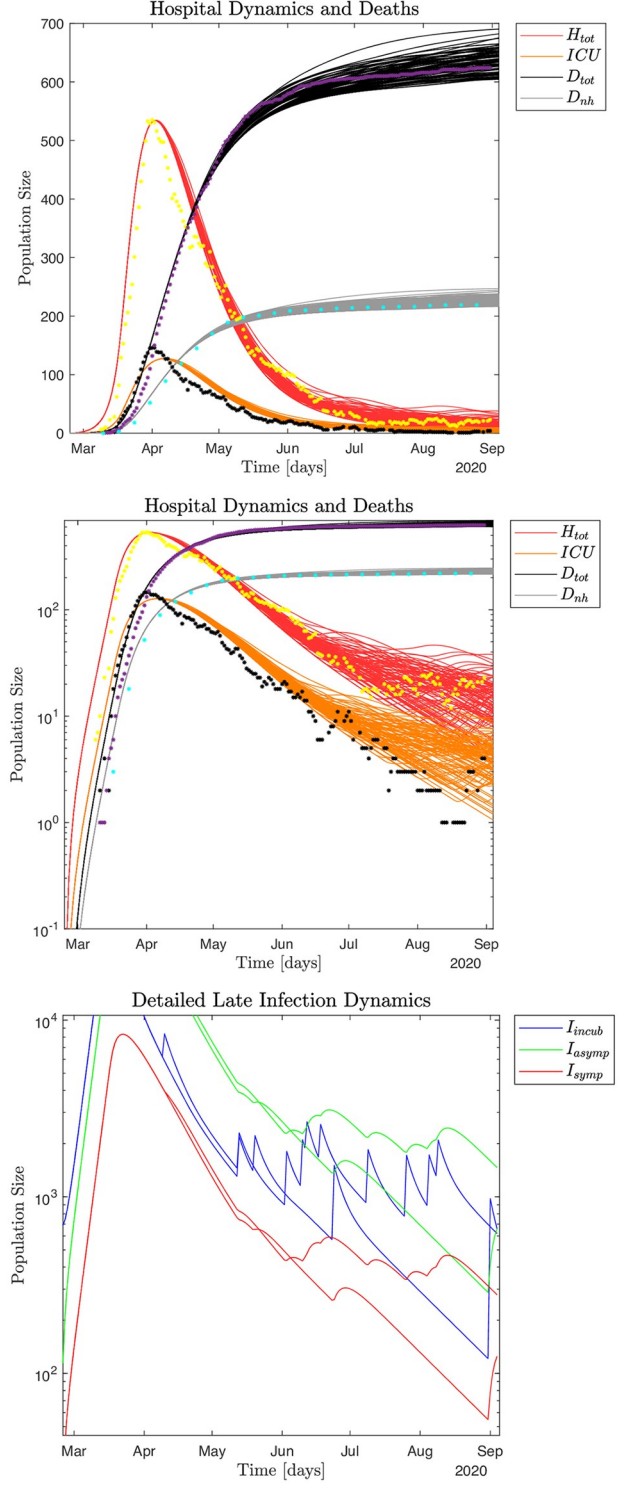

**Fig 7.**

he or she is sick and will likely seek medical advice as well as self-isolate, while an asymptomatic individual likely would continue his or her daily routines at home, go to school or go to work as well as engage in social activities. In some model studies asymptomatic and weakly symptomatic are treated separately [17].

Because historical total hospital $H_{tot}$ and *ICU* occupations as well as death data only stem from the symptomatic infected population $I_s$, and we assume a constant fraction of seriously ill individuals are in need of hospitalization [11], the symptomatic population size is at any given time constrained by these real life data. The same is true about the non-hospital deaths. In contrast, the asymptomatic population could in principle vary freely if we did not include additional empirical information. We may use the Danish serological test study for SARS-CoV-2 antibody prevalence per May 28, 2020, to scale the pandemic: i.e., to estimate the relative scale of the symptomatic versus the asymptomatic populations. For details see S1 File. Since we find the prevalence to be about 1.37% (~ 79,700 recovered individuals), we get $\rho_s \simeq 0.16$ that means approximately 16% of the infected are symptomatic while 84% of the infected are asymptomatic; thus ~ 5.2 times more asymptomatic than symptomatic infected individuals. Note that as Denmark only had one published prevalence study per August 31, 2020, our estimated scale of the pandemic significantly relies on this one measurement.

## Iso-symptomatic infection diagram

The parameters (i) $\zeta$: the relative infectiousness of asymptomatic versus symptomatic; (ii) $\rho_s$: the relative fraction of infected (incubated) that becomes symptomatic; (iii) $\beta_s^0$: the initial infection parameter (and the initial $\Re_0$, that depends on the three previous parameters) define critical properties of COVID-19 and other communicable infectious diseases. Our goal is obviously to identify the most realistic combinations of these critical parameters so that we can reproduce the historical hospital and death data and at the same time satisfy existing clinical knowledge about these parameters. However, *a priori* and without introduction of extra data, infinitely many, although only very particular, combinations of $\zeta$, $\rho_s$ and $\beta_s^0$ can reproduce the historical hospital and death data. It is this very particular relationship between $\zeta$, $\beta_s^0$, $\rho_s$, and $\Re_0$ we explore in Fig 8 as it defines some deeper characteristics of the COVID-19 pandemic.

Fig 8 shows the relationship that exists between $\zeta$, $\beta_s^0$, $\Re_0$, and $\rho_s$, which we may call an *iso-symptomatic infection diagram*. We assume the infection parameters $\zeta$, $\rho_s$, and $\beta_s$ are constant over time and throughout a free (non-restricted) pandemic. Thus $\beta_s = \beta_s^0$ together with the initial $\Re_0$ define for the initial free pandemic period. Note, that we have estimated the actual ($\zeta$, $\beta_s^0$, $\rho_s$ $\Re_0$) values for the Danish epidemic. This in part from clinical studies that defines $\zeta$ [14] and in part from the antibody prevalence study for blood donors 18–65 years in late May 2020 together with PCR testing which includes children <18 years until end of August 2020 [8]. These additional data together with our SEIRS model define $\beta_s^0$ (and $\Re_0$) and $\rho_s$. The iso-symptomatic infection diagram is constructed from multiple Monte Carlo—Least Square parameter optimized simulations that all match the observed Danish COVID-19 hospital occupation data at the onset of the pandemic. In this construction we use the more generic infection model with variable $\zeta$ as defined in Eqs (3)–(6). For more details, please see S1 File.

## Estimation of mortality rates of the Danish COVID-19 epidemic

The estimated Infection Fatality Rate (IFR) as of August 31, 2020, based on the simulations that include asymptomatic cases, can be calculated by dividing the total number of deaths by the total number of recovered: 628/173,544 = 0.0036 or ~ 0.4%.

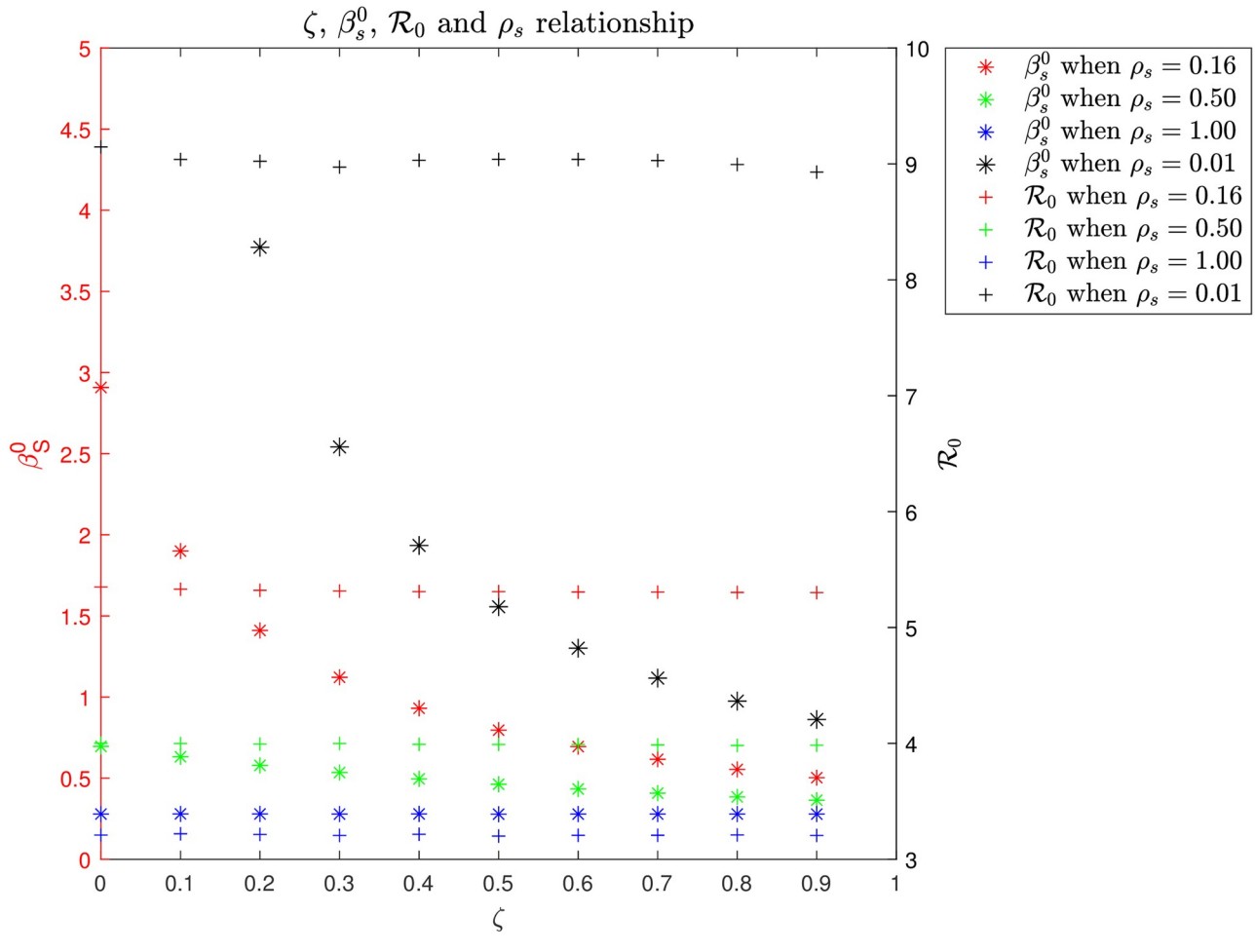

**Fig 8.**

The official calculations of mortality of COVID-19 in Denmark in the beginning of the pandemic were based on hospitalized patients. The value was found to be 3.3% April 3, 2020, just before Easter, and 4.8% June 19, 2020, when the testing strategy had been changed for about 2 months to also include non-hospitalized individuals. However, asymptomatic cases of COVID-19 were not included in these calculations and therefore the numbers were Case Fatality Rates (CFR). Thus, according to our calculations the true mortality rate of COVID-19 in Denmark has been overestimated by a factor of approximately 10.

In another location of the Danish Kingdom, the Faeroe Islands (52,484 inhabitants), testing was far more extensive from the beginning of the epidemic (109,233 tests as of September 14, 2020) to detect and isolate *all* contacts. They have recorded 423 COVID-19 cases (0.8% including asymptomatic cases) and no deaths (both IFR and CFR) (0% - 95% c.l. 0.0–0.9%) [8]. Note that our estimated mortality of 0.4% in Denmark is in the middle of their confidence interval.

In Iceland (364,260 inhabitants), a serological study estimated that 0.9% of the population had been infected with SARS-CoV-2 and that the IFR was 0.3%, which is again similar to our estimates [19].

**Table 2. Estimated infected population sizes and infection rates as a function of hospital population using Eqs.**

**Hospital & Infected Populations**

| Hospital population | Infected population | Daily infected | Observable population |
|---|---|---|---|
| 10 | 1,053 | 70 | 807 |
| 20 | 2,106 | 140 | 1,615 |
| 50 | 5,265 | 351 | 4,036 |
| 100 | 10,530 | 702 | 8,073 |
| 250 | 26,324 | 1,755 | 20,182 |
| 500 | 52,649 | 3,501 | 40,364 |
| 1,000 | 105,297 | 7,020 | 80,728 |
| 6,000 | 631,784 | 42,119 | 484,367 |
| 15,000 | 1,579,459 | 105,297 | 1,210,918 |
| 30,000 | 3,158,918 | 210,595 | 2,421,837 |
| 60,000 | 6,317,835 | 421,189 | 4,843,674 |

(S32)—(S38). Note the linear proportionality between the numbers; e.g., 10 times higher hospital population yields 10 times higher infected populations and infection rates. See text for details.

## Quasi steady state approximation

For extended periods the hospital occupation may be approximately constant indicating a steady state situation of the pandemic. For example, we saw that for the period July—August, 2020. If we assume a steady state pandemic situation we approximately have:

$$\frac{dI_i}{dt} = \frac{dI_a}{dt} = \frac{dI_s}{dt} = \frac{dH_{tot}}{dt} = 0 \tag{11}$$

We can then use our model to estimate the sizes of the infected background populations $I_i$, $I_a$, and $I_s$ as we can calculate backwards from a known approximately constant hospital occupation. In a steady state situation we have $h_{frac}\gamma_s I_s - (1-\rho_{h,icu})\gamma_h H - (1-\rho_{icu,h})\gamma_{icu} ICU = 0$ that expresses: what goes into the hospital system equals what leaves the hospital system. With known parameters $h_{frac}$, $\gamma_s$, $\rho_{h,icu}$, $\gamma_h$ and known $H$ and $ICU$ populations the size of $I_s$ can be estimated. Then $I_a = ((1-\rho_s)/\rho_s)I_s$ and $I_i = (I_a + I_s)/2$, as the average incubation time is half the average infection time of both the symptomatic and asymptomatic. See S1 File for further details. The key population numbers for different $H_{tot}$ population sizes are estimated and tabulated in Table 2 based on the steady state approximation (all numbers are calculated on a computer with double precision and then rounded to integer values).

Since $h_{frac}$ is a critical parameter both for these steady state studies and our simulations, we investigate the impact of varying this value around the standard value of 0.082 or 8.2%. Please see S1 File for this analysis.

## Testing efficiencies

From mid-July to mid-August, 2020 the nationwide voluntary testing program together with medically recommended testing and contact tracing [20]. in average identify and isolate $T_d \sim$ 85 infected/day from the contagious populations. By comparing the number of identified infected $T_d$ with the number of estimated contagious individuals $I_{obs}$ gives a simple indicator of the testing efficiency. We may define a single day detection efficiency as the detected number within a day divided with the estimated number of infected on that same day: $DT_{abs} = T_d/I_{obs} = 85/1615 = 0.0526$ or approximately 5.3%. So, in a theoretical situation where every

individual in the country could be tested in a single day, all observably infected would be identified in one day with a total efficiency of 100%.

However, from an operational point of view a more relevant efficiency number is $DT_{daily}$ the net daily detected and isolated compared to the daily newly infected defined $I_{new}$. Obviously, the impact of the detection and isolation critically depends on which time point in an infected individual's disease process it occurs. Since an explicit modeling of the infection and contact tracing processes is outside of the scope of our current study, we will only provide upper and lower bounds for the impact of the infection and contact tracing processes.

The impact would be maximal if the detection and isolation of all individuals occur during the incubation period and before the incubating infected become contagious, i.e., in the first 3.5 days of the incubation period. Thus, we can calculate an upper bound of the impact of the net daily detection and isolation efficiency as $DT_{daily}^{max} = T_d/I_{new} \simeq 85/140 = 0.6071$ or approximately 61%.

The impact would be minimal if the detection and isolation of all individuals are asymptomatic infected occurs the very last day of their time as infectiousness. However, we need to recall that throughout this study we have assumed that almost all the symptomatic infected are detected and isolated in Denmark, so part of the daily identification and isolation process involves the symptomatic infected. Since $I_s$ = 225 in the steady state, we can deduce that the epidemic produces $I_s\gamma_s$ = 22.5 new symptomatic infected each day that are assumed detected. So, 85 detections/day minus 22.5 detections/day = 62.5 detections/day are available for detection from $I_a$ at their last contagious day. In our model there are 11.5 days of infectiousness for both the symptomatic and asymptomatic populations: 1.5 days of the incubation times plus 10 days of the symptomatic or asymptomatic time. If for simplicity we assume equal contagiousness every day we get that the symptomatic can only infect 1.5 days during the incubation time as we assume they are detected and isolated once they become symptomatic. The asymptomatic can infect 1.5 days during the incubation time and 9 days during the asymptomatic infection period, because they are only detected on the 10th and last day of this period.

Thus, we get a lower bound on the net daily detection efficiency

$$DT_{daily}^{min} \simeq \left(\frac{11.5-10.5}{11.5} \times 62.5 + \frac{11.5-1.5}{11.5} \times 22.5\right)/140 = 0.1786 \text{ or approximately 18.0\%.}$$

We now have estimates for a lower and upper bound on the daily testing efficiency for the period July 15 and August 15, 2020 given as 18% < $DT_{daily}$ < 61%. As we have no detailed knowledge about the details of the infection and contact tracing processes, we may use the average (center point) of the upper and the lower bound as a rough estimate for the daily testing efficiency: (18 + 61) / 2 $\simeq$ 39.5 or approximately 40%. Obviously, more accurate estimates could be calculated from an explicit modeling and simulation of the involved infection and contact tracing activities.

Similarly, for the period October 1–20, 2020 (see S1 File for details on the estimated infected population sizes), we obtain a single day detection efficiency $DT_{abs} = T_d/I_{obs}$ = 407/9,351 = 0.0435 or approximately 4.4%. $DT_{daily}^{max} = T_d/I_{new} \simeq 407/831 = 0.501$ or approximately 50% while $DT_{daily}^{min} \simeq \left[\frac{11.5-10.5}{11.5} \times \left(\frac{1301}{10}\right) + \left[\frac{11.5-1.5}{11.5} \times (407 - 130)\right]/813 \simeq 0.3102$ or approximately 31%. Thus, a rough estimate for the daily testing efficiency is (50.1 + 31.02) /2 = 40.55 or approximately 41% that within rounding errors is the same daily testing efficiency as over the summer.

There are two important lessons from the above estimates:

(i) The testing, contact tracing, and isolation program in Denmark both provides a key indicator for the current geographic infection trends, but also takes part in balancing the background infection increase. This is particularly true when geographically focused efforts are

undertaken. Removing and isolating on average 85 infected per day July 15—August 15, 2020 together with behavioral restrictions had a significant impact in controlling the epidemic. Without the testing and the resulting daily removal of infected individuals the Danish pandemic would likely have been in an expansion phase during this period.

(ii) The Danish testing program, which is among the most ambitious in the world, has a testing efficiency of approximately 40% for both periods investigated, while the single day infection tracing efficiency indicator is found to be 5.3% mid July to mid August 2020 and 4.4% for October 1–20, 2020.

## Hospitalized and estimated infected populations in other regions: Spain and USA

If we assume the biology of the COVID-19 pandemic is similar in Denmark, Spain and the US, and we further assume that the hospital system including *ICU*s in these countries at least qualitatively function in a similar manner regarding treatment of COVID-19 patients, we may scale the Danish hospitalization numbers and the corresponding infected population estimates to larger populations. Obviously more detailed studies are necessary to verify—or falsify—the validity of such a comparison, but as a simplified first approximation we believe such estimates provide appropriate order of magnitude estimates.

In Table 2 we list estimates for infected and observed population sizes based on the total hospitalized population size and the quasi steady state approximations given in Eqs. (S32) to (S38). For the estimates we assume the number of intensive care unit patients are approximately one fifth of every patient, thus $ICU = 0.2 \times H_{tot}$ ($k = 0.8$ in Eq. (S38)).

Potential healthcare differences across different countries can be adjusted by adjusting the appropriate healthcare system parameters in Eq. (S32).

By the end of August, 2020, Spain had about 6,000 hospitalized with COVID-19 [21], which according to our estimates corresponds to a total infected population of about 630,000 with about 484,000 observable and about 42,000 newly infected per day. In late August Spain detected and isolated a little less than 10,000 positive COVID-19 cases/day [22]. Thus, the Spanish single day testing efficiency indicator is approximately $10,000/484,000 \simeq 0.0207$ or 2.1%.

To estimate the net daily detection efficiency, we need to approximate $(DT_{daily}^{max} + DT_{daily}^{min})/2$. $DT_{daily}^{max} = T_d/I_{new} \simeq 10,000/42,000 = 0.2381$ or approximately 24%. To calculate $DT_{daily}^{min}$ we note that $I_s \simeq 67,390$, so $I_s\gamma_s \simeq 6,739$ new symptomatic infected are produced every day that we assume are detected. Since $T_d \simeq 10,000$, 10,000–6,739 detections are thus available for individuals in $I_i$ and $I_a$, recall discussion in the last subsection. $DT_{daily}^{min} \simeq \left(\frac{11.5-10.5}{11.5} \times 6,739 + \frac{11.5-1.5}{11.5} \times (10,000 - 6,739)\right)/42,119 \simeq 0.0812$ or approximately 8.1%. This gives an approximate net daily detection efficiency of (23.81 + 8.12)/2 = 15.965 or approximately 16% by the end of August 2020.

By the end of September, 2020, the US had two hospitalization peaks, both with about 60,000 patients, one in mid- April and one in late July 2020 [23], as well as two valleys, both with about 30,000 patients, one in mid-June and one mid- September 2020. The corresponding estimates from Table 2 indicate that 30,000 hospitalizations correspond to about 3.16 mil. infected while 60,000 hospitalizations correspond to about 6.32 mil. infected. The daily detection average for the US during September 2020 was about 40,000 while the daily newly infected average was about 210,000. Thus, the single day testing efficiency in the US during September was about $40,000/2,420,000 \simeq 0.0165$ or 1.7%.

To estimate the net daily detection efficiency, we need to approximate $(DT_{daily}^{max} + DT_{daily}^{min})/2$. $DT_{daily}^{max} = T_d/I_{new} \simeq 40,000/210,595 = 0.1899$ or approximately 19%. To calculate $DT_{daily}^{min}$ we note that $I_s \simeq 336,951$, so $I_s\gamma_s \simeq 33,695$ new symptomatic infected are produced every day that we assume are detected. This assumption seems less appropriate with the actual numbers involved as almost all of the detections would then stem from symptomatic infected. However, to keep consistency in the estimations we keep this assumption. Since $T_d \simeq 40,000$, 40,000–33,695 detections are thus available for individuals in $I_i$ and $I_a$, recall discussion in last subsection. $DT_{daily}^{min} \simeq \left(\frac{11.5-10.5}{11.5} \times 33,695 + \frac{11.5-1.5}{11.5} \times (40,000 - 33,695)\right)/210,595 \simeq 0.0399$ or approximately 4%. This gives an approximate net daily detection efficiency of (18.99 + 3.99)/ 2 = 11.49 or approximately 11.5% during September 2020.

The above estimates from Spain and the US should only be viewed as an illustration of the method developed in Eqs. (S32)—(S38) applied to other countries. Healthcare differences across different countries should be implemented by adjusting the appropriate healthcare system parameters in Eq. (S32) as well as our assumption about detection and isolation of symptomatic infected.

## "What if?" scenarios

**Best case scenario: Continued micro-outbreak scenarios with $\Re_0 \simeq 1$.**   Since April 2020 Denmark has experienced a number of localized micro-outbreaks that were contained due to isolation, contact tracing, and increasingly more extensive testing. A narrative discussion of these micro-outbreaks was given in the Introduction, while in the Results Section we discuss how we model and simulate the impact of such micro-outbreaks by means of added internal noise.

Previously we used noise to trigger injections of newly infected into the population to generate local micro-outbreaks. Local fluctuations in the behavior from time to time triggers micro-outbreaks probably in part caused by a combination of events or places where many people are gathered and super-spreaders are present. In these situations the general background dynamics is characterized by $\Re_0$ slightly smaller than 1.0 so that the background dynamics and the noise induced micro-outbreaks together generate a close to steady state infection level that corresponds to what we may call a macroscopic effective $\Re_0 \simeq 1$. We interpret the general background dynamics as the sum of the behavior of the general population together with the testing and contact tracing activities that result in a daily removal of infected from the epidemic, which together defines a $\Re_0$ that is smaller than one.

It should be emphasized that there is a critical difference between injecting additional infected into the general population versus changing the behavior in the general population. If the background infection is decreasing, $\Re_0 < 1$, an injection of newly infected has the same impact whether they e.g., are returning travelers that were infected elsewhere or they stem from a local micro-outbreak from within the country. In an $\Re_0 > 1$ situation both effects add equally to an already expanding pandemic.

From August 14–31, 2020, Denmark adjusted the reopening of the country. The most significant changes included opening of higher educational and vocational institutions as well as longer opening hours of restaurants and bars. However, the restaurant and bar openings were partly rolled back in the first part of September together with a number of additional restrictions due to observed increase in infection levels and hospitalizations [18].

If, as a "best case scenario" we assume no changes in the general human behavior compared to the summer of 2020, as well as no impact by the cooler fall and winter temperatures, we obtain the results shown in Fig 9. We still assume recurring localized micro-outbreaks with the

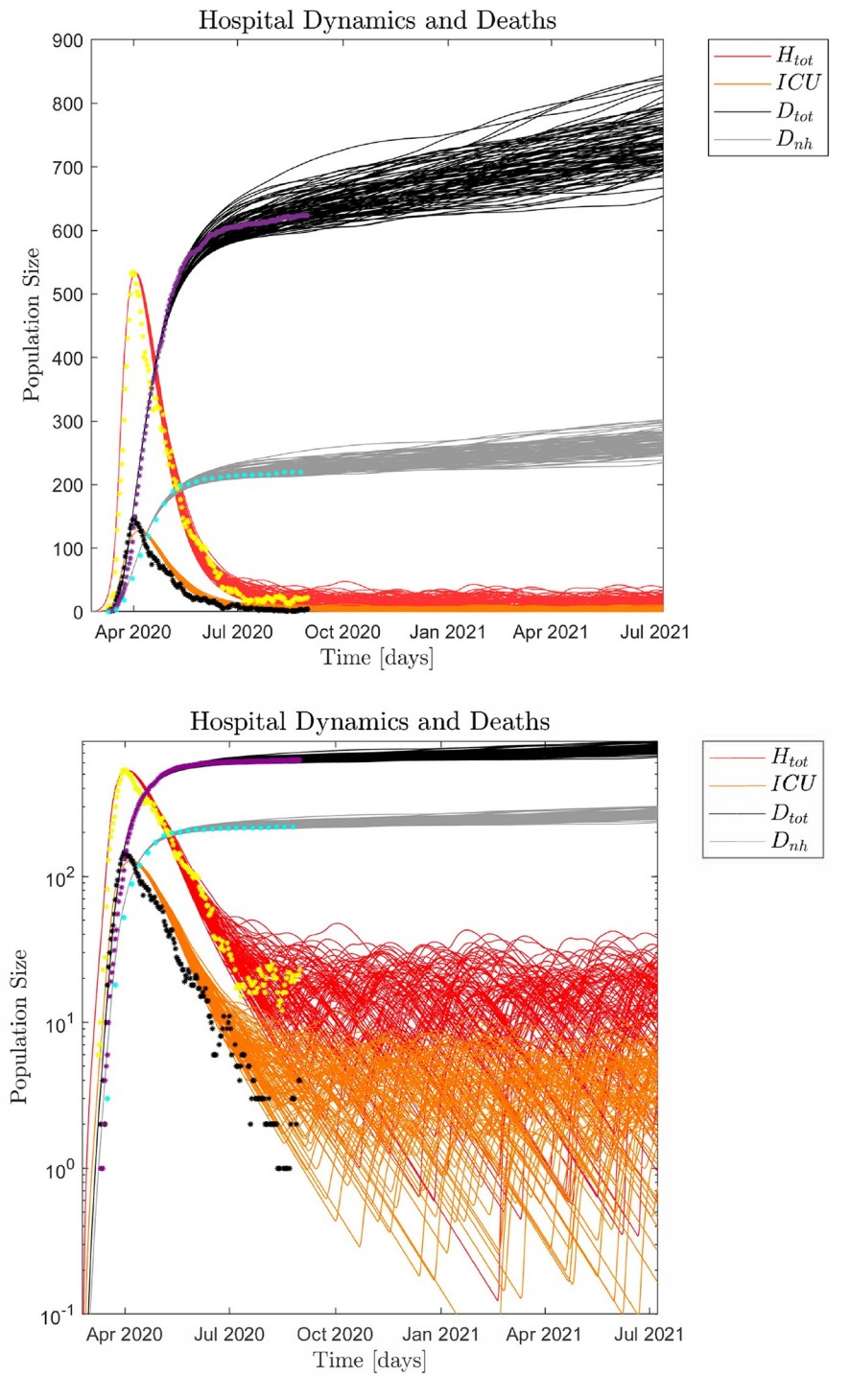

**Fig 9.**

same frequency and amplitude as previously had, which is an optimal dynamical regime to operate in until a vaccine becomes available.

Since the pandemic dynamics are operating around $\Re_0 \sim 1$, a quasi steady state can occur for any hospital occupation level. A low hospital occupation is obviously the most desirable.

**Aggressive infection scenario following the seasonal influenza pattern.**    If all behavioral and policy restrictions are lifted without a universally available vaccine, a new aggressive infection wave would immediately emerge. If we further assume that we still keep the symptomatic infected effectively isolated, i.e. $\beta_s = 0.01\beta_s^0$, we get a scenario where the pandemic is governed by a free infection spread almost solely by the asymptomatic $I_a$ and incubating $I_i$ populations. Such a scenario yields (Eqs (7)–(10)),

$$\mathfrak{R}_0 = \frac{\beta_i^0}{\gamma_i} + \frac{(1 - \rho_s)\beta_a^0}{\gamma_a} + \frac{\rho_s 0.01\beta_s^0}{\gamma_s}$$

$$= \frac{\beta_s^0(\rho_s + (1 - \rho_s)\zeta)0.351}{\gamma_i} + \frac{(1 - \rho_s)0.309\beta_s^0}{\gamma_a} + \frac{\rho_s 0.01\beta_s^0}{\gamma_s} \tag{12}$$

$$= 3.004 \sim 3.0 \tag{13}$$

Both the initial growth in the $H_{tot}$ observations as well as the simulations indicate that the initial pandemic February—March 2020 with $\mathfrak{R}_0 \sim 5.4$ had a doubling time of ~ 2.4 days while a pandemic with $\mathfrak{R}_0 \sim 3.0$ would have a doubling time of ~ 4.2 days. Thus, as long as we effectively isolate the symptomatic infected a worst case scenario of such an infection wave cannot be as aggressive as the initial wave we experienced early 2020. Thus, future infection waves would be expected to emerge with $\mathfrak{R}_0 \leq 3.0$ and a doubling time of $\geq 4.2$ days.

The most vulnerable period for an aggressive Danish COVID-19 infection wave would likely be at the time the country usually experiences it's yearly influenza epidemic (Fig 14 in S1 File).

Fig 10 shows an aggressive scenario with $\mathfrak{R}_0 = 3.0$ and $\beta_s = 0.01\beta_s^0$, which means that 99% of all symptomatic infected are isolated, where the scenario is adapted to the yearly seasonal epidemic influenza pattern. To explore the full impact of such an infection, we further assume no policy interventions for asymptomatic and incubating individuals, which means that incubating and asymptomatic individuals move freely to interact with and potentially infect the susceptible population. In particular, note that due to the assumed limited immunological memory for the asymptomatic infected ($1/\xi_a = 60$ days; decay time) the pandemic does not "burn out" as the susceptible population is continuously replenished by the recovered population.

Without intervention the pandemic peaks for $I_i$ December 30, 2020 and January 7, 2021 for $I_a$ and $I_s$, and by February 7, 2021, approximately 4,071,000 people (~ 70% of the population) would have been infected and recovered from the pandemic. The simulated hospital impact peaks around January 17, 2021 with ~ 19,680 hospitalized and January 20, 2021 with $\simeq$ 4,786 in *ICU*s which would completely overwhelm the Danish healthcare system. It is estimated that the Danish *ICU* capacity is about 1,000 beds and the total hospital capacity is about 15,000 beds. A year after the onset of this infection wave (November 1, 2021) about 56,690 people would have died from the SARS-CoV-2 virus, which is about the same as the total number of deaths in Denmark in the year 2019: 53,958 [24].

**Free pandemic: Worst case scenario.**    "What would have happened if no policy interventions were imposed at the beginning of the pandemic?" In Fig 11 data are generated in simulation using the standard parameters also used in Figs 4 and 5, but assuming no behavioral modifications or policies imposed at the onset of the pandemic, thus $\mathfrak{R}_0 \simeq 5.36$. This is a hypothetical scenario where the pandemic rages freely.

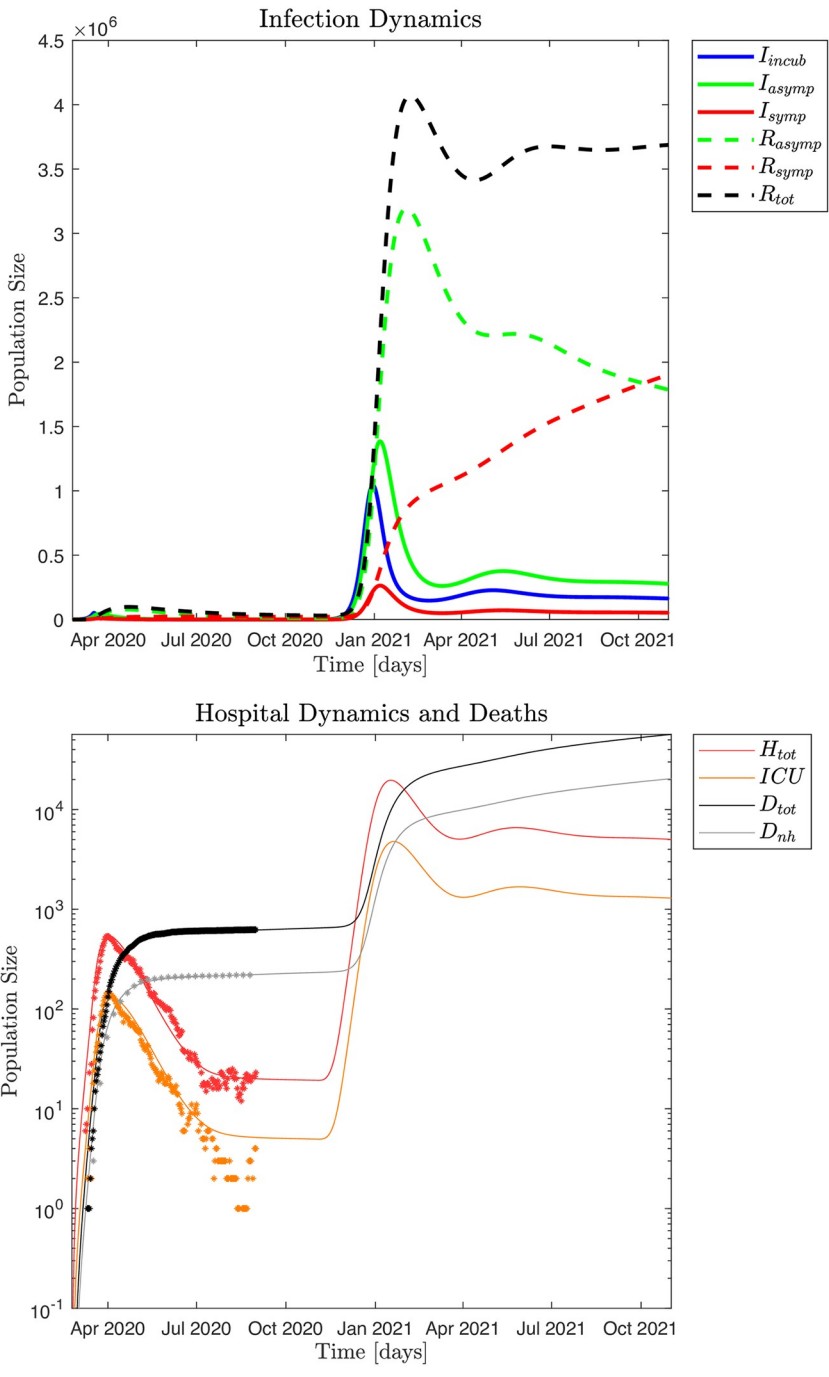

**Fig 10.**

The pandemic does not "burn out" in this scenario either, as the susceptible population is continuously replenished by the recovered population (Fig 11). This means that a relatively stable hospital population of about $H_{tot} \simeq 6,500$ by October 2020 would slowly decrease to $\simeq$ 4,500 in October 2021. A conservative estimate of the expected excess deaths from such a free pandemic is $\simeq 64,960$ after one year, by February 24, 2021, which is more than all the deaths in Denmark in 2019 (53,958) [24].

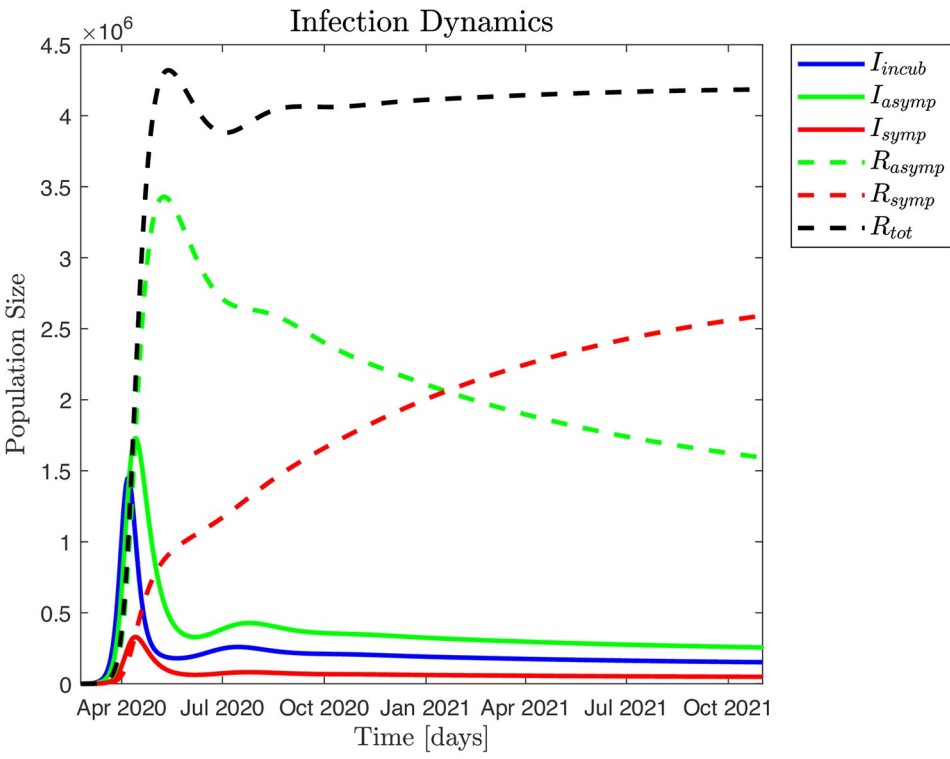

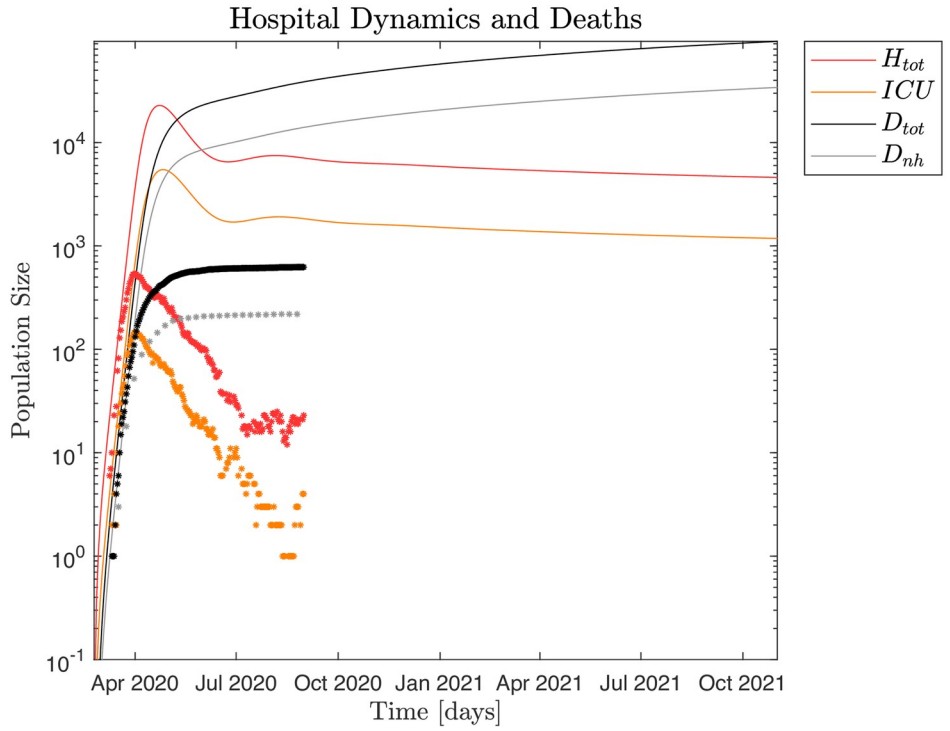

**Fig 11.**

## Discussion

The goal of this work is to provide a theoretical platform to help understand the details of the Danish COVID- 19 epidemic that can hopefully also provide useful insights about the nature of the pandemic in other countries and regions. Our models and simulations can easily be adapted to other regions or countries where hospital, *ICU* and death data are available.

We have chosen a simple macroscopic description and used a differential equation-based simulation of the pandemic and health care system dynamics that seems appropriate for most of the issues we seek to address, although a microscopic agent-based simulation scheme would have been more appropriate e.g. to capture the dynamics of the observed micro-outbreaks. Our approach is also limited in two other aspects. We neither have a geographic representation of the pandemic nor do we have an age disaggregated population although reported data clearly show features caused both by geography and age differences in the population.

However, by adding noise to our model and conducting Monte Carlo simulations of large ensembles of the system we can to some extent compensate for the shortcomings caused by the lack of geographic representations and the occurrence of localized micro-outbreaks. This approach is further supported because Denmark is a small country with a relatively high mobility and population mixing. The missing age distribution does not cause serious problems for understanding the initial epidemic dynamics, but for the later micro-outbreaks we in principle need to know the nature of their age distribution as this significantly impacts the health care system. We have not implemented such parameter changes in our simulations. All parameters are kept constant over time in each of our simulations except the externally imposed lockdown and reopening policies/behaviors. By adjusting the amplitude and frequency of the added noise we can adjust the expected impact on the health care system from such micro-outbreaks.

The micro-outbreaks since late July 2020 began in a slaughterhouse in the city of Ringsted. Then came the outbreaks in the three largest cities in Denmark: first in Århus and then around September 1, 2020 in Odense and Copenhagen. These outbreaks were driven by young people aged 20–29 years and immigrants from Middle East and Somalia who live in rather closed communities. The outbreaks spread to other groups, but fewer old and vulnerable persons became infected compared to the epidemic in the spring. The number of patients hospitalized, in particular patients treated in *ICU*s and the number of deaths, remained very low in the early and mid-fall period compared to the spring epidemic although it was still the old people who dominated the hospitalizations and deaths. In each of the micro-outbreaks non-pharmacological measures were re-introduced and the $\Re_t$ of each micro-outbreak went from above 1.5 to approximately 0.8 within a month, which probably is to be expected considering the epidemic dynamics of COVID-19.

The testing capacity during these outbreaks went up to nearly 1% of the whole Danish population each day and by October 7, 43% of the population had been tested for SARS-CoV-2. This means that many persons who contacted SARS-CoV-2 positive persons including asymptomatic persons were detected. Therefore, of the 9,623 SARS-CoV-2 positive persons in the fall outbreak only 34 (0.37%) died, whereas 434 (4.8%) of the 8,851 SARS-CoV-2 positive patients who were hospitalized during the spring epidemic died. The reduced mortality is similar to an case-fatality rate of 0.4% in a recent report from Hong Kong [25]. The early high rate mainly reflects that only patients who were hospitalized were tested for SARS-CoV-2 in the spring epidemic and is therefore CFR. The IFR of the spring epidemic could not be calculated and, therefore, the true mortality was estimated too high. This is important to realize, because a restricted testing capacity may mislead the health-care system, the population and the precautions taken to manage the epidemic, e.g., based on the calculated $\Re_0$ and $\Re_t$. The burden on

the hospitals and *ICU*s was much lower in the outbreaks in the fall. The spring/fall ratio of the maximum number of normal hospital unit COVID-19 patients was 535/147 while the corresponding ratio for *ICU* patients was 153/20. The improved treatment of COVID-19 patients with e.g. Remdesevir, dexamethasone, and prophylaxis and treatment of bleeding disorders [26, 27], is probably the reason for the proportionally lower number of hospitalized patients who in the fall were treated in *ICU*s or died.

The estimated percentage of symptomatic infected that needs hospital care $h_{frac}$ is a critical parameter as it scales the connection between the main observables, the time series for $H$ and *ICU* occupations, the death toll, and the infection dynamics. We have used an age aggregated estimate for $h_{frac}$ = 8.2% based on the age distribution of health care needs from [11]. With an increase of $h_{frac}$ to 13.9% the simulation still corresponds reasonably well to the observed data we have access to, while a decrease in $h_{frac}$ to 6.1% does not allow the simulation to both satisfy the time series of our main observables and the measured seroprevalence of May 28, 2020. See S1 File for details on this analysis.

For our hospital model we have also chosen simplicity over including more details that potentially could help capture the health care dynamics in greater detail. The simple hospital model is nonetheless able to reproduce both the historical hospital and death data quite well, but it has difficulties reproducing some of the reported data details. For example, both the reported $H_{tot}$ and *ICU* occupation time series have sharper peaks at the onset of the pandemic than the simulated peaks. Also to enable the simulated *ICU* data to follow the reported time series data in a reasonable manner we need to use an average *ICU* occupation time $1/\gamma_{icu}$ = 10 days, which is shorter than the reported average occupation time in the spring of 13 days, although it should also be noted that a more recent improved treatment of COVID-19 patients means that fewer have a severe course [26, 27].

Requiring our healthcare model to reproduce the historical hospital $H_{tot}$ and *ICU* data, the simulation underestimates the total accumulated number of hospital patients $H_{tot}$ by 7–14% and it also underestimates the accumulated number of *ICU* patients. Thus, our simple health care model has difficulties simultaneously matching both the time series data (daily hospital occupation numbers) and the accumulated data (total number of hospitalized patients).

There is an ongoing discussion of the best value for average period of infectiousness both for the symptomatic and asymptomatic populations, where (1) suggests $1/\gamma_s$ = $1/\gamma_a$ = 10 days while (30) suggests 7 days as a better estimate. Lowering the infectious period $1/\gamma_s$ and $1/\gamma_a$ both for the symptomatic and the asymptomatic populations from 10 to 8 days in our simulation makes it difficult to reproduce the total hospital occupation numbers (too fast a decline of simulation numbers after lockdown), while for the simulated death numbers from the hospitals (too fast a rise). However, shorter $1/\gamma_s$ and $1/\gamma_a$ values make it easier to reproduce the initial relatively sharp peak for the *ICU* occupation number. To compensate for the lower $1/\gamma_a$ and $1/\gamma_s$ values the $\beta$ values must increase a bit. With lower infection times minor adjustments are needed for the parameters associated with hospitals, *ICU*s, and death tolls.

We can use MC optimization of the parameters where we minimize the LS difference between the reported and the simulated time series of the total hospital occupation ($H$ + *ICU*). Despite the minor mismatches between what the MC-LS parameter optimization yields and what visual parameter inspection yields, the MC-LS capability enables us to do large scale explorations of the parameter impacts.

Using a steady state approximation for the background infection that is indicated by approximately constant low values in our main observables, i.e., $H$ and *ICU* occupations and death data, enables us to make simple analytical estimates of the sizes of the corresponding infected populations $I_i$, $I_a$ and $I_s$. These analytical estimates are robust to most parameter perturbations and they are confirmed by numerical simulations using steady state conditions.

This approach and the corresponding infected population estimates should give more accurate predictions of the actual sizes of the infected populations than direct infection testing of the daily scale currently being done.

Another general finding coming out of this study is the tight relationship that exists between $\beta_s$, $\Re_0$, $\rho_s$ and $\zeta$, which can be represented by an iso-symptomatic-infection diagram, that is made possible by using MC-LS optimization for each $\beta_s$, $\Re_0$, $\rho_s$ and $\zeta$ parameter combination. We propose such diagrams could readily be constructed for most infectious diseases, so that one diagram would hold crucial and comparable quantitative information about critical parameters that define the dynamic characteristics of a pandemic.

## Conclusions

Our computational platform explores the connection between the dynamics of the epidemic, the national health care system and the imposed policies.

The relative frequency between symptomatic and asymptomatic infected was 16% and 84% respectively. The $\Re_0$ was ~ 5.4 for the initial free pandemic, $\Re_0$ ~ 0.4 for the lockdown period, and $\Re_0$ ~ 0.8–1 for most of the successive reopening periods with short bursts where $\Re_0 > 1$, especially in localized areas.

Over the summer Denmark has been operating with $\Re_0$ ~ 1 from June through August 2020. The estimated infected population sizes for the quasi steady state period mid-July to mid-August, 2020 are $I_i \simeq 702$; $I_a \simeq 1{,}179$; $I_s \simeq 225$; $I_{tot} \simeq 2{,}106$; and $I_{obs} \simeq 1{,}615$, while the daily infection rate is $I_{new} \simeq 140$.

The daily net testing efficiency was approximately 40% for the periods July 15—August 15, 2020 and October 1–20, 2020. The single day identification efficiency indicator in these periods, defined as the number of positively identified infected over the estimated observable infected population the same day, is about 5%.

Since the symptomatic infected are effectively isolated from the population, and if behavioral policies are kept in place and observed, future COVID-19 infection waves fortunately would not be as aggressive as the first wave assuming the SARS-CoV-2 pathogenity does not change significantly due to mutations. A new infection wave will have a lower bound with doubling time of more than 4.2 days and less than $\Re_0$ ~ 3.0 compared to the initial wave with a doubling time of about 2.4 days and $\Re_0$ ~ 5.4.

Our simulation platform is suitable for exploring forecasting scenarios for the upcoming winter and beyond or until large scale vaccination becomes available. Our results indicate that it is possible to more precisely than previously [28, 29] to calculate the burden on the hospitals and *ICU*s of new COVID-19 infection waves or a new pandemic. Thereby, our simulation might help minimize the impact on hospital treatment of non-COVID-19 patients.

We have shown that the mortality (IFR) of COVID-19 in Denmark is only 0.4% like the IFR in Iceland, the Faeroe Islands and the CFR in Hong Kong. Our simulation indicates that had Denmark not adopted any behavioral measures to counteract the Danish pandemic the death toll would be $\simeq 64{,}960$ by February 24, 2021 one year into the pandemic. That surpasses the total number of deaths (53,958) in Denmark in the year 2019 [24]. This simulation based COVID-19 death estimate is conservative, as we have assumed a fully functional healthcare system that can scale up to the needed capacity. The free pandemic in such a scenario would have resulted in a hospital $H_{tot}$ and *ICU* occupation peaks in late April, 2020 that would have completely overwhelmed the Danish health care system with $\simeq 22{,}850$ hospitalized and $\simeq 5{,}472$ in *ICU*s. Per January 1, 2021 there would still be $\simeq 5{,}860$ hospitalized and $\simeq 1{,}500$ in *ICU*s in a free COVID-19 pandemic according to such a simulation scenario.

Clearly Denmark's adopted behavioral modifications, lockdown, measured reopening, testing, and contact tracing procedures, have been highly successful in mitigating the epidemic from the onset through August 2020.

## Supporting information

**S1 File.**
(DOCX)

## Acknowledgments

We are grateful for the general guidance of John Erik Hansen regarding all epidemic matters in the early stages of this study, as well as a final critical review of the manuscript. We are also grateful for the enlightening conversations with Gitte Kronborg regarding the COVID-19 situation at Danish hospitals in the early stages of the epidemic. Oana Ciofu and Asbjørn Dahl are acknowledged for their critical review of the manuscript and we thank Hans Ziock for his in depth critical reading, corrections, and constructive suggestions throughout the last versions of the manuscript. All remaining errors in the manuscript are the responsibility of the authors.

## Author Contributions

**Conceptualization:** Steen Rasmussen.

**Data curation:** Niels Høiby.

**Formal analysis:** Steen Rasmussen, Michael Skytte Petersen.

**Investigation:** Steen Rasmussen, Michael Skytte Petersen, Niels Høiby.

**Methodology:** Steen Rasmussen, Michael Skytte Petersen, Niels Høiby.

**Project administration:** Steen Rasmussen.

**Software:** Steen Rasmussen, Michael Skytte Petersen.

**Supervision:** Steen Rasmussen, Niels Høiby.

**Validation:** Steen Rasmussen, Michael Skytte Petersen, Niels Høiby.

**Visualization:** Steen Rasmussen, Michael Skytte Petersen.

**Writing – original draft:** Steen Rasmussen.

**Writing – review & editing:** Steen Rasmussen, Michael Skytte Petersen, Niels Høiby.

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
