## [Decision Letter · Decision Letter 0]

2 Feb 2021

PONE-D-20-39535

SARS-CoV-2 infection dynamics in Denmark, February through October 2020: Nature of the past epidemic and how it may develop in the future

PLOS ONE

Dear Dr. Rasmussen,

Thank you for submitting your manuscript to PLOS ONE. After careful consideration, we feel that it has merit but does not fully meet PLOS ONE’s publication criteria as it currently stands. Therefore, we invite you to submit a revised version of the manuscript that addresses the points raised during the review process.

Please review comments made by the reviewers and provide point by point response in your revised manuscript

We look forward to receiving your revised manuscript.

Kind regards,

Muhammad Adrish

Academic Editor

PLOS ONE

Journal Requirements:

2.Thank you for stating the following financial disclosure:

 "The funders had no role in study design, data collection and analysis, decision to pu"

3. Please include your tables as part of your main manuscript and remove the individual files. Please note that supplementary tables (should remain/ be uploaded) as separate "supporting information" files.

Reviewers' comments:

Reviewer's Responses to Questions

**Comments to the Author**

1. Is the manuscript technically sound, and do the data support the conclusions?

Reviewer #1: Yes

Reviewer #2: Yes

2. Has the statistical analysis been performed appropriately and rigorously? 

Reviewer #1: Yes

Reviewer #2: Yes

3. Have the authors made all data underlying the findings in their manuscript fully available?

Reviewer #1: Yes

Reviewer #2: Yes

4. Is the manuscript presented in an intelligible fashion and written in standard English?

Reviewer #1: Yes

Reviewer #2: Yes

5. Review Comments to the Author

Reviewer #1: This is a mathematical study related to the behavior of the SARS-CoV-2 pandemic in Denmark, from the first cases in February to October 2020. The study included the SEIRS model (susceptible, exposed, infected, recovered, susceptible), and calculated some variables including the infection mortality rate, the transmission rate, in patients symptomatic and asymptomatic.

Likewise, the description focuses on a single European country with a low population density, where the behavior of the epidemic has been clearly different comparing with other countries, however, the authors consider that the model can easily be used in other scenarios, and in other countries with different population and infection rates.

A very favorable point in this study is the presence of a static pandemic, a pandemic with exponential growth and a pandemic with mini-outbreaks.

The study itself uses a highly specialized mathematical language that makes it difficult to adjust to the language normally used by non-mathematic readers. The different parts of the manuscript are too long: introduction, methods, results and conclusions. It is very difficult to read (again, for non-mathematics and related areas involved). Maybe data could be used in other scenarios with different population rates, but I’m not sure if could be used in others health care systems. I consider it is an excellent manuscript, but most readers will not be mathematicians, so reading can be very difficult. I suggest a magazine with specialized readers in this area.

Reviewer #2: The study aimed to provide a theoretical platform to help understand the details of the Danish COVID- 19 epidemic. Despite some limitations, this is a well designed and written study which can provide useful insights about the nature of the pandemic in other countries and regions

6. PLOS authors have the option to publish the peer review history of their article (what does this mean?). If published, this will include your full peer review and any attached files.

Reviewer #1: No

Reviewer #2: No

---

## [Author Response · Author response to Decision Letter 0]

8 Mar 2021

We have included a letter with Response to Reviwers where all reviewer comments are addressed.

---

## [Decision Letter · Decision Letter 1]

24 Mar 2021

SARS-CoV-2 infection dynamics in Denmark, February through October 2020: Nature of the past epidemic and how it may develop in the future

PONE-D-20-39535R1

Dear Dr. Rasmussen,

We’re pleased to inform you that your manuscript has been judged scientifically suitable for publication and will be formally accepted for publication once it meets all outstanding technical requirements.

Kind regards,

Muhammad Adrish, MD, MBA, FCCP, FCCM

Academic Editor

PLOS ONE

Additional Editor Comments (optional):

All comments/suggestions have been addressed by the authors.

Reviewers' comments:

Reviewer's Responses to Questions

**Comments to the Author**

1. If the authors have adequately addressed your comments raised in a previous round of review and you feel that this manuscript is now acceptable for publication, you may indicate that here to bypass the “Comments to the Author” section, enter your conflict of interest statement in the “Confidential to Editor” section, and submit your "Accept" recommendation.

Reviewer #1: All comments have been addressed

2. Is the manuscript technically sound, and do the data support the conclusions?

Reviewer #1: Yes

3. Has the statistical analysis been performed appropriately and rigorously? 

Reviewer #1: Yes

4. Have the authors made all data underlying the findings in their manuscript fully available?

Reviewer #1: Yes

5. Is the manuscript presented in an intelligible fashion and written in standard English?

Reviewer #1: Yes

6. Review Comments to the Author

Reviewer #1: This is a pandemic model to predict the impact of COVID-19 on the healthcare system in Denmark classifying patients symptomatic and asymptomatic. The model based in susceptible-exposed-infected-recovered- susceptible included deaths inside and outside the hospital, and made the estimations based in seroprevalence, including mini-outbreaks presented in that country. They calculated case fatality rate and infection mortality rate, the transmission rate during the initial free epidemic period, during lock down and in the reopening periods. They found low fatality rate because major people were asymptomatic or few symptoms. This epidemic model system can be extrapolated to other countries with different populations sizes and to other seasons and when prevalence be higher.

The manuscript has several figures which help to understand the results, and improve the reading.

The main changes comparing with the previous version are the results were rewritten leaving the mathematical formulas essential to continue reading, without having an excess in these, and facilitating reading for the reader not specialized in mathematics. The methods and the results were also summarized.

7. PLOS authors have the option to publish the peer review history of their article (what does this mean?). If published, this will include your full peer review and any attached files.

Reviewer #1: No

---

## [Editor Report · Acceptance letter]

31 Mar 2021

PONE-D-20-39535R1 

SARS-CoV-2 infection dynamics in Denmark, February through October 2020: Nature of the past epidemic and how it may develop in the future 

Dear Dr. Rasmussen:

I'm pleased to inform you that your manuscript has been deemed suitable for publication in PLOS ONE. Congratulations! Your manuscript is now with our production department. 

Kind regards, 

on behalf of

Dr. Muhammad Adrish 

Academic Editor

PLOS ONE